# Auditory perception and neural representation of temporal features are altered by age but not by cochlear synaptopathy

Friederike Steenken[1,2,3†], Rainer Beutelmann[1,2,3†], Henning Oetjen[1,2,3], Christine Köppl[1,2,3], Georg M Klump[1,2,3]*

[1]Cluster of Excellence 'Hearing4all,' Carl von Ossietzky University Oldenburg, Oldenburg, Germany; [2]Research Centre Neurosensory Science, Carl von Ossietzky University Oldenburg, Oldenburg, Germany; [3]Department of Neuroscience, School of Medicine and Health Science, Carl von Ossietzky University Oldenburg, Oldenburg, Germany

*For correspondence:
georg.klump@uni-oldenburg.de

†These authors contributed equally to this work

## eLife Assessment

This study tested the specific hypothesis that age-related changes to hearing involve a partial loss of synapse connections between sensory cells in the ear and the nerve fibers that carry information about sounds to the brain, and that this interferes with the ability to discriminate rapid temporal fluctuations in sounds. Physiological, behavioral, and histological analyses provide a powerful combination to test this hypothesis in gerbils. Contrary to previous suggestions, it was found that chemically-induced isolated synaptopathy (at similar levels as observed in aged gerbils) did not result in worse performance on a behavioral task measuring sensitivity to temporal fine-structure, nor did it produce degradations in auditory-nerve fiber encoding of fine structure. Aged gerbils showed degraded behavior and stronger than normal envelope responses, but temporal fine-structure coding was not affected; interpreted by the authors as suggesting central processing contributions to aging effects on discrimination. These findings are **important** for advancing our knowledge of the mechanistic bases for age-related changes to hearing, and the evidence provided is **solid** with the results largely supporting the claims made and minor limitations related to possible confounds discussed in reasonable depth.

**Abstract** Age-related hearing loss is a complex phenomenon. The earliest-onset degenerative event is the gradual loss of neural connections between the cochlea and auditory brainstem. To probe for perceptual deficits that might arise from this loss, cochlear synaptopathy was induced pharmacologically in young-adult gerbils, which were then tested in a challenging listening task for the perception of temporal fine structure. Treated gerbils behaved no differently than normal-hearing, young-adult animals. In contrast, old gerbils, which typically express many cochlear and central-neural pathologies, showed impaired perception. To probe for the underlying mechanisms, single-unit responses were obtained from the auditory nerve to the same test stimuli. Responses from old gerbils showed no impairment in temporal locking to the stimulus fine structure. However, responses were significantly more driven by slower temporal fluctuations of the stimulus envelope, suggesting that the central auditory system may be unable to extract the relevant information for discrimination from such altered inputs.

## Introduction

Many elderly people have difficulties understanding speech in noisy environments, despite having normal tone-detection thresholds in quiet. To explain this condition of compromised supra-threshold perception, for example, of speech in background noise (e.g. *Füllgrabe et al., 2014*; *Gómez-Álvarez et al., 2023*; *Lopez-Poveda, 2014*; *Moore, 2014*; *Parthasarathy et al., 2020*), despite an absence of detectable hearing deficits in the pure-tone audiogram in quiet, is currently an important challenge in auditory neuroscience. It has been suggested to be associated with a reduced number of synapses between auditory-nerve (AN) fibers and inner hair cells (IHCs), which is commonly referred to as cochlear synaptopathy (e.g. *Bharadwaj et al., 2014*; *Kujawa and Liberman, 2009*; *Liberman, 2017*; *Liberman and Kujawa, 2017*). Synaptopathy is one of the aging processes in the cochlea, and, in addition, can be caused by excessive noise exposure, both accumulating during a human's lifetime (e.g. *Liberman et al., 2016*; *Wu et al., 2019*, but see *Johannesen et al., 2019*). Animal experiments demonstrate that exposure to loud sounds can produce such synaptopathy, even without permanent loss in hearing sensitivity (e.g. *Kujawa and Liberman, 2009*). Furthermore, the number of functional synapses decreases with increasing age (e.g. gerbil: *Bovee et al., 2024*; *Gleich et al., 2016*; *Steenken et al., 2021*; mouse: *Jeng et al., 2020*, *Parthasarathy and Kujawa, 2018*, *Sergeyenko et al., 2013*; rat: *Cai et al., 2018*; human: *Viana et al., 2015*).

Synaptopathy is postulated to be causal for compromised processing of temporal stimulus features (e.g. *Parthasarathy and Kujawa, 2018*), especially of the temporal fine structure (TFS) of sounds (reviewed in *Moore, 2019*), resulting in difficulties of speech comprehension in background noise. Specific psychophysical tests that evaluate the performance to distinguish between stimuli that differ in TFS have been developed. These tasks are then expected to be sensitive to synaptopathy (for review, see *Moore, 2014*). The most commonly used test is the TFS1 test (*Moore and Sek, 2009*), which probes the discrimination of harmonic and inharmonic tone complexes that strongly differ in TFS but only marginally in the envelope of the waveform. The previously found link between diminished perception of stimuli differing in TFS (*Moore et al., 2006*) and sensory-neural hearing loss in human patients has also been found in TFS1 test results (*Mathew et al., 2016*). Furthermore, a correlation between age and TFS1 sensitivity has been observed (e.g. *Eipert et al., 2019*; *Füllgrabe et al., 2014*; *Moore et al., 2012*).

Whereas post-mortem studies of human subjects have provided evidence for age-related synapse loss (*Viana et al., 2015*; *Wu et al., 2019*), it has been impossible to relate this loss to deficits in supra-threshold sound perception of living subjects. Animal studies have suggested a close correlation between the number of surviving synapses and supra-threshold amplitude growth of auditory brainstem responses (ABRs) or compound action potentials (CAPs; e.g. *Bourien et al., 2014*; *Bramhall et al., 2018*; *Buran et al., 2010*; *Kujawa and Liberman, 2009*; *Lin et al., 2011*; *Sergeyenko et al., 2013*; *Shi et al., 2016*). A number of attempts have been made to infer the synapse loss in humans from non-invasive physiological measures, such as ABR, and psychophysical measures of the processing of sounds. However, even studies involving large samples of human subjects have not produced strong evidence that synaptopathy in humans has functional consequences for supra-threshold temporal perception (e.g. *Bramhall et al., 2019*; *Carcagno and Plack, 2022*; *Guest et al., 2018*). Attempts to relate lifetime noise exposure of human subjects to their supra-threshold perceptual deficits have also not been conclusive (e.g. *Prendergast et al., 2017*; *Prendergast et al., 2019*; *Füllgrabe et al., 2020*). Controlled animal studies are one strategy for testing the influence of synapse loss on discrimination of complex sounds that differ in TFS. In those, synaptopathy can be experimentally manipulated and the degree of cochlear dysfunction can be related to neural responses and performance in behavioral tests, using the same stimuli. With this approach, we can test the hypothesis that synaptopathy negatively affects the discrimination of stimuli that differ in their TFS.

Animal studies relating behavioral measures of performance in response to a variety of stimuli to synaptopathy have so far also provided mixed results (reviewed in *Henry, 2022*). An investigation by *Henry and Abrams, 2021*, which experimentally induced synaptopathy in budgerigars by kainic acid infusions at the round window, failed to demonstrate a relationship between the amplitude of the CAP (taken as a measure of cochlear synaptopathy) and tone-detection thresholds in noise. Studies in Macaque monkeys that produced a noise-induced hearing loss by acoustic trauma demonstrated differences between normal-hearing and acoustically traumatized monkeys for (a) tone detection in noise (*Burton et al., 2020*), (b) masking release for tone detection in modulated maskers compared to

unmodulated maskers, and (c) spatial release from masking for tone detection in noise (*Mackey et al., 2021*). However, these studies used high trauma levels and long exposure durations. Thus, effects were not limited to synapse loss at the IHC but included both outer hair cell (OHC) and IHC damage. Studies in chinchilla AN fibers, involving normal-hearing subjects and subjects with noise-induced hearing loss, did reveal differences in the neural envelope representation but failed to demonstrate differences in the neural representation of TFS that match the psychophysical effects observed in human subjects (e.g. *Henry and Heinz, 2012*; *Kale and Heinz, 2010*; *Kale et al., 2014*).

To clarify the relation between synaptopathy, AN fiber responses, and behavior, we embarked on a study in Mongolian gerbils (*Meriones unguiculatus*) that combined behavioral measurements of sensitivity in the TFS1 test and the representation of harmonic and inharmonic complex tones with the same fundamental frequency ($f_0$) by AN fibers. In addition, synaptopathy of many of the experimental subjects was characterized by directly counting the numbers of functional synapses on IHCs, and these measures were related to sensitivity in the TFS1 test. Because compromised performance in the TFS1 test has been observed in elderly humans, we compared the perception of young-adult and old gerbils. In order to experimentally induce synaptopathy in young-adult gerbils and thus dissociate the effects of synaptopathy from effects of age, we treated young-adult gerbils with an infusion of ouabain at the round window. This treatment has been shown to reduce the number of synapses on gerbil IHCs (*Bourien et al., 2014*). We observed a lower perceptual sensitivity of old compared to young-adult gerbils in the TFS1 test. However, contrary to our hypothesis, ouabain treatment did not affect the performance in the TFS1 task, and the responses of AN fibers that survived ouabain treatment did not differ from those of gerbils that were not treated.

## Results
### Synapse loss was similar in old gerbils and gerbils treated with a high dose of ouabain

Synapses were counted at cochlear locations corresponding to 2, 4, 8, and 16 kHz in 54 gerbils, divided into the five gerbil groups (*Table 1*). The most important region for this study is the one corresponding to 2 kHz, because it matches best with the TFS1 stimuli. The other frequencies are reported for comparison and completeness. Young-adult, untreated gerbils had, on average, 19–23 synapses per IHC, depending on the cochlear location (*Figure 1*). Synapse numbers were significantly different between gerbil groups (*Table 1*). Compared to untreated young adults, the average number of synapses per IHC in gerbils treated with 70 μM ouabain was significantly reduced at all cochlear locations, to between 12 and 19. Similarly, in old gerbils, the reduction compared to young adults was significant for all locations up to 8 kHz, with average numbers of synapses per IHC between 15 and 19 (*Table 1*). No significant difference between ears treated with a high dose of ouabain and old ears was found, indicating that both animal groups had similar degrees of synaptopathy at all cochlear locations evaluated. In surgery-only ears and ears treated with 40 μM ouabain, average synapse numbers were not significantly different from untreated young-adult ears, suggesting no effect of the surgical procedure itself and no effect of the lower ouabain dose. Based on this, we

**Table 1.** Synapse number was reduced after ouabain treatment and in old age, at all cochlear locations evaluated.
The mean number of functional synapses per inner hair cell (IHC) was significantly different between gerbil groups for cochlear locations corresponding to all four frequencies (univariate ANOVAs: 2 kHz: $F=5.496$, $p=9.782 \times 10^{-4}$; 4 kHz: $F=4.995$, $p=1.988 \times 10^{-3}$; 8 kHz: $F=7.933$, $p=5.974 \times 10^{-5}$; 16 kHz: $F=6.176$, $p=4.305 \times 10^{-4}$). Post-hoc tests revealed significant differences (at the 5% level, with Bonferroni correction) between the young-adult and the ouabain-high groups for all equivalent frequencies, between the young-adult and the old group for 2, 4, and 8 kHz, and between the ouabain-low and the ouabain-high group for 16 kHz.

|  | 2 kHz | 4 kHz | 8 kHz | 16 kHz |
| --- | --- | --- | --- | --- |
| Young-adult | 23 (+/-0,5, N=19) | 21 (+/-0,4, N=18) | 19 (+/-0,5, N=18) | 20 (+/-0,5, N=18) |
| Surgery only | 23 (+/-0,5, N=3) | 18 (+/-1, N=3) | 16 (+/-0,4, N=3) | 21 (+/-0,4, N=3) |
| Ouabain low | 22 (+/-0,7, N=8) | 19 (+/-0,6, N=8) | 16 (+/-0,3, N=8) | 19 (+/-0,6, N=8) |
| Ouabain high | 19 (+/-1,4, N=10) | 17 (+/-1,4, N=10) | 12 (+/-1,7, N=10) | 13 (+/-2,5, N=11) |
| Old | 19 (+/-0,7, N=14) | 17 (+/-0,8, N=12) | 15 (+/-0,6, N=12) | 16 (+/-0,6, N=13) |

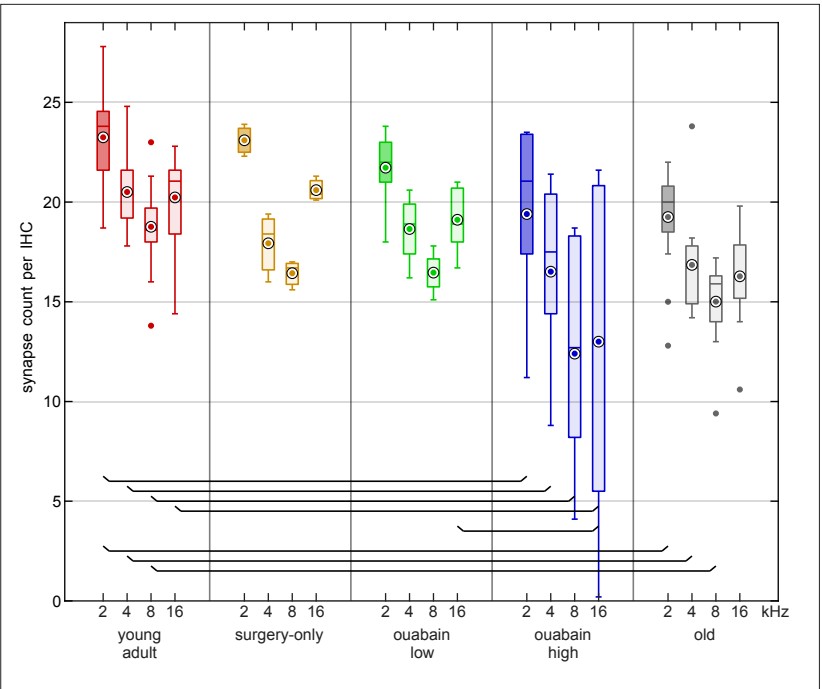

**Figure 1.** Synapse number was reduced after ouabain treatment and in old age. Number of functional synapses per inner hair cell (IHC), as a function of cochlear position (expressed as corresponding frequency), for young-adult gerbils (red), surgery-only-treated gerbils (orange), gerbils treated with a low dose (40 µM) of ouabain (green), gerbils treated with a high dose (70 µM) of ouabain (blue), and old gerbils (gray). Box plots display the median (center line), mean (white circled dot), 25th and 75th percentiles (upper and lower edges of the boxes), and maximum and minimum (whiskers). Boxes are highlighted for the cochlear position corresponding to 2 kHz, the region that was the best match to the TFS1 stimuli. Brackets show significant differences in post-hoc tests. The corresponding statistics are detailed in the legend of *Table 1*.

pooled the single-unit data from surgery-only and 40 µM ouabain-treated gerbils into a 'sham' group. The gerbils treated with 70 µM ouabain are henceforth referred to simply as 'ouabain.' There were no significant differences in synapse numbers between gerbils tested behaviorally and the other gerbils from the corresponding groups not tested behaviorally (two-way ANOVA: factor inclusion in behavior [$F_{(1,44)}=0.330$, $p=0.568$] and no interactions between inclusion and gerbil groups [$F_{(4,44)}=1.179$, $p=0.333$]).

**Table 2.** Numbers of gerbils of different treatment and age groups within the behavioral and electrophysiological part of this study. Note that numbers in categories 2 and 3, plus the numbers evaluated for synapse histology (listed in *Table 1*) do not sum up to total numbers, since many gerbils were used in all three study parts. For the neural data, the core data for this study are the auditory-nerve (AN) fibers, which were tested with TFS1 stimuli. However, some additional fibers, in some animal groups also additional animals, were included for the analysis of neural tuning; these numbers are provided separately. Furthermore, both untreated gerbil groups were supplemented with data from eight young-adult gerbils and twelve old gerbils for which synapse counts were available, from *Steenken et al., 2021*.

| | | Young adult | Surgery only | Ouabain 40 µM | Ouabain 70 µM | Old |
|---|---|---|---|---|---|---|
| 1 | **Total** number of animals in this study | 21 | 4 | 8 | 10 | 17 |
| 2 | Animals in **behavioral** study | 4 | 4 | 5 | 8 | 3 |
| 3 | Animals with **neural tuning** data (# AN fibers) neural TFS1 resp. (# AN fibers) | 5 (27) 5 (37) | 3 (17) 2 (10) | 6 (38) 3 (8) | 7 (21) 6 (14) | 6 (13) 6 (23) |

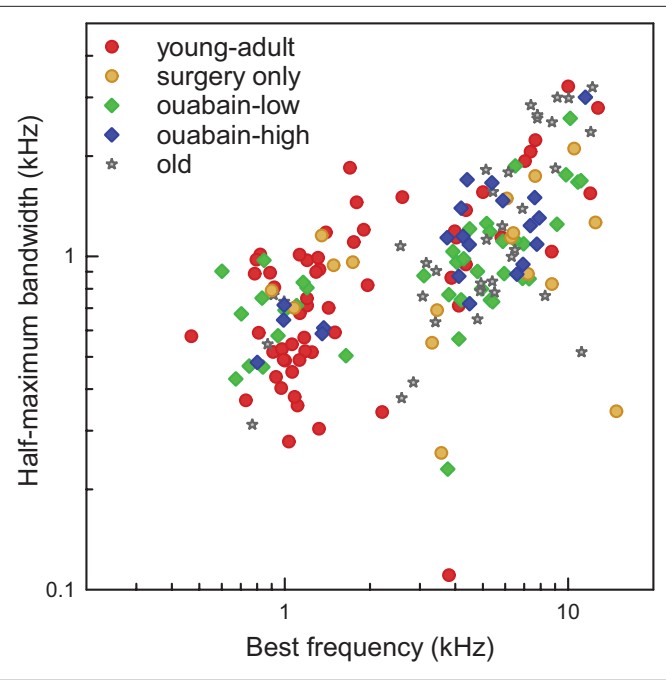

**Figure 2.** Bandwidths of neural response curves were not affected by ouabain treatment or age. Half-maximal bandwidth of response curves of single auditory-nerve (AN) fibers, obtained with pure tones at a level ≤20 dB above the individual fibers' rate threshold, as a function of best frequency (BF). Color code for the different treatment groups is the same as in *Figure 1*.

## Threshold sensitivity and frequency tuning were not affected by the synapse loss

Induced synaptopathy, with a similar dose of ouabain, and causing a similar extent of synapse loss as in the present study, was previously found not to impair threshold sensitivity (assessed via ABR) or OHC function (assessed via distortion-product otoacoustic emissions; *Bourien et al., 2014*). Here, we assessed CAP thresholds and single-unit AN tuning to test for normal, basic cochlear function in our young-adult, ouabain-treated gerbils (see *Table 2* for sample sizes of AN fibers). CAP thresholds in response to chirp stimuli were obtained from all ouabain- and sham-treated animals before surgery, and again at the start of terminal neurophysiology, and in most cases, for both ears. None of our treatment groups showed a significant threshold difference between the two measurements (Mixed-model ANOVA, [F(2,35)=0.222, $p$=0.802]; mean difference <5 dB). Neural bandwidths of single AN fibers clearly varied with BF (*Figure 2*), as expected, but did not differ significantly between untreated young adults and gerbils treated with 70 μM ouabain (Mixed-model ANOVA, [BF effect: F(1,76) = 71.67, $p$=1.41 × 10$^{-12}$], [treatment effect: F(1,76) = 0.228, $p$=0.634]). Thus, the ouabain treatment did neither affect basic cochlear sensitivity nor frequency selectivity.

In our old gerbils, CAP thresholds showed the typical, variable sensitivity loss observed previously for individuals of comparable age (*Heeringa, 2024*). The mean threshold elevation was 36 dB, compared to young adults (T-test, $p$=1.77 × 10$^{-5}$, n(young-adult cochleae)=38, n(old cochleae)=15). It was also shown previously that old gerbils still show normal, sharp frequency selectivity of their tuning curve tips, i.e., near threshold (*Heeringa et al., 2020*; *Hellstrom and Schmiedt, 1996*). This was confirmed for our sample of old gerbils (*Figure 2*; Mixed-model ANOVA, [BF effect: F(1,90) = 101.58, $p$=1.96 × 10$^{-16}$], [age effect: F(1,90) = 2.28, $p$=0.135]) and is consistent with only mild OHC pathologies in aging gerbils (*Tarnowski et al., 1991*; *Adams and Schulte, 1997*; *Steenken et al., 2024*).

## Average rate did not carry sufficient information about inharmonic frequency shift

Spike responses to TFS1 stimuli of 92 AN fibers from 22 animals were recorded (for details, see *Table 2*). BFs ranged from 470 Hz to 11,528 Hz, with a median of 3995 Hz. The SRs of these fibers

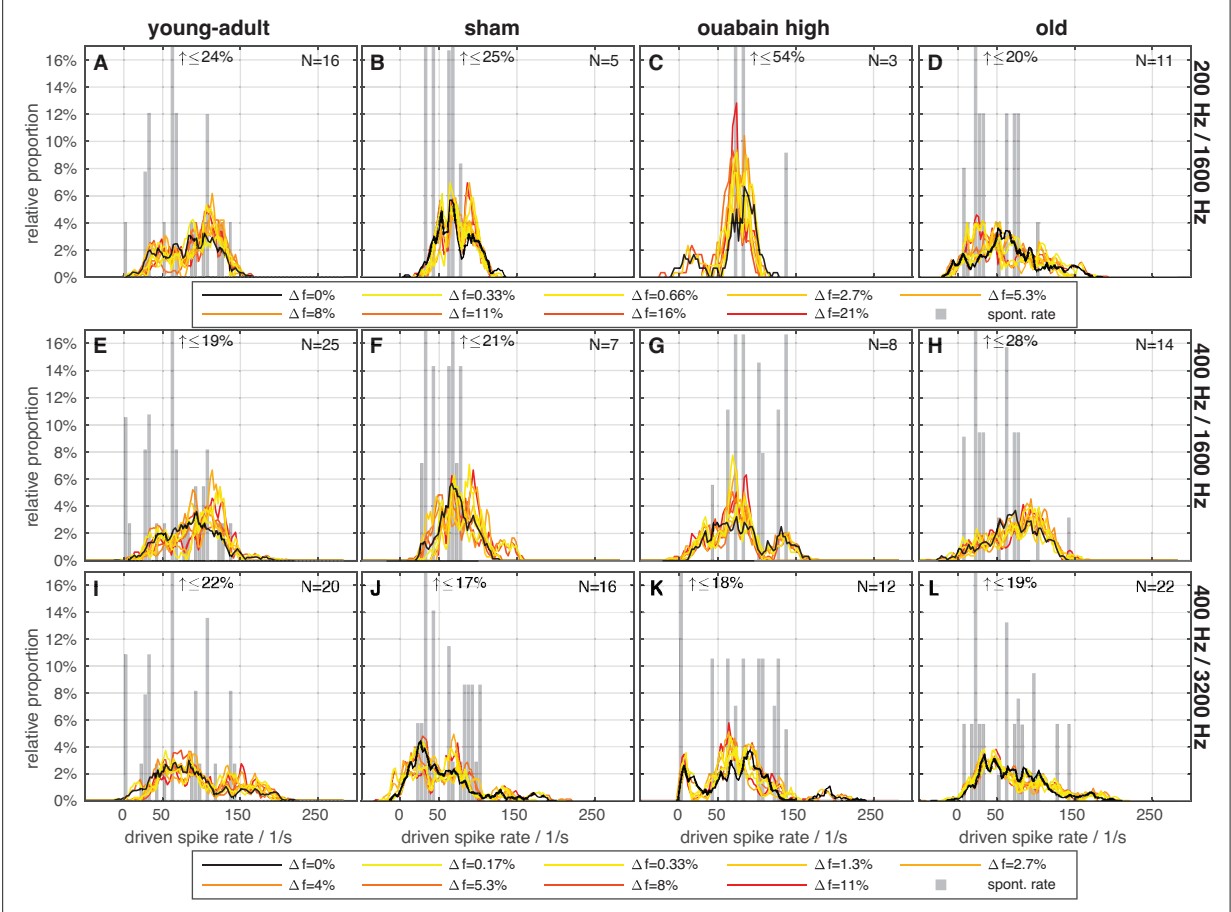

**Figure 3.** Mean-driven spike rates did not differ between frequency shifts. Histograms of average driven spike rates during presentation of the TFS1 stimuli (black and colored) and average spontaneous rate, without acoustic stimulation (gray). Subject groups are arranged in columns and stimulus conditions in rows. Colors code the frequency shift Δf in percent of f₀, with the responses to harmonic stimuli shown in black and responses to the inharmonic stimuli grading from yellow (least inharmonic) to red (most inharmonic). The legends apply to panels above them. In each panel, the maximal number of fibers (with 40 stimulus repetitions per fiber) that the histograms are based on is indicated; note that not all fibers were tested with each frequency shift Δf. Clipped spontaneous rate bars are marked with an arrow and the maximal relative proportion beyond the axis limit.

ranged from 0 sp/s to 144 sp/s with a median of 60 sp/s. Average stimulated rates (4.8 sp/s - 223 sp/s, median 135 sp/s) and driven rates (i.e. average rate minus SR, 4.8 sp/s - 215 sp/s, median 67.6 sp/s) were calculated from the spike counts across the stimulus duration of 375 ms, excluding on- and offset ramps.

Most importantly, the inharmonic frequency shift Δf% - as a covariate in the ANOVA - had no significant effect on mean driven rate [F(1,943)=0.018, p=0.8933] (**Figure 3**, overlapping colored and black distribution curves in each panel). The overall distribution of driven rate also did not change substantially with Δf%. Thus, the inharmonic frequency shift had no effect on the mean driven rate, and AN spike rate was not a potential cue for the gerbil in the behavioral task.

Beyond this robust, basic result, there were differences in discharge rates between the animal groups and stimulus conditions that might reflect different sampling biases of single units in the different experimental groups (see **Figure 4** and further detailed statistics results reported in Appendix 1). We acknowledge that our single-unit sample was not sufficient, i.e., not all subpopulations of AN fibers were sufficiently sampled for all animal treatment groups, to conclusively resolve every aspect that may be interesting to explore. The power of our approach lies in the direct linkage of several levels of investigation – cochlear synaptic morphology, single-unit representation, and behavioral performance – and, in the main manuscript, we focus on the core question of synaptopathy and its relation to TFS perception.

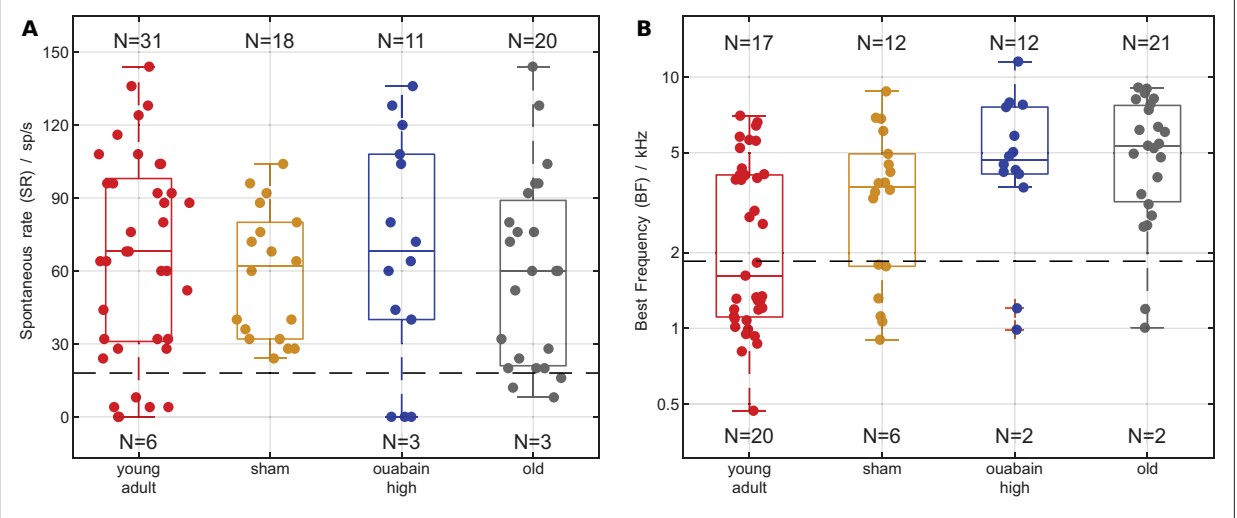

**Figure 4.** Exploring sampling biases in the neural data. (**A**) Median spontaneous rates (SR) did not differ between animal groups. Individual spontaneous rates (SR) for all auditory-nerve (AN) fibers reported in this study (dots), group medians (box center lines), quartiles (box boundaries), minima, and maxima (whiskers). The dashed black line indicates the boundary between low- and high-spontaneous-rate fibers (18 spikes/s). Counts of included fibers within each combination of group and spontaneous rate (SR) class are indicated at the corresponding side of the box plots. A Kruskal-Wallis test confirmed that the group medians are not significantly different at the 5% level. (**B**) Median best frequency (BF) was lower in the young-adult sample. Individual best frequencies for all auditory-nerve fibers reported in this study (dots), and box plots, in the same style as in **A**. The dashed black line indicates the boundary between low and high best frequency (BF) class (1.85 kHz), as used in the results statistics. Counts of included fibers within each combination of group and best frequency class are indicated at the corresponding side of the box plots. A Kruskal-Wallis test confirmed that the group medians are significantly different at the 5% level, with a Bonferroni-corrected post-hoc test showing that only the young-adult group median differs significantly from all other groups.

## Neural representation of TFS was not degraded in AN fibers of old or synaptopathic gerbils

The same set of AN fibers was probed for the neural responses to the fine structure and envelope of TFS1 stimuli (for details see *Table 2* in Materials and methods) using the TFS log-z-ratio (see Methods section 'Neural recordings - Data analysis'). A high TFS log-z-ratio indicates strong phase locking to the stimulus TFS. The main findings are listed below; for more details, see Appendix 1. Importantly, the inharmonic frequency shift, Δf%, as a covariate in the ANOVA, affected the TFS log-z-ratio significantly and strongly [main effect: $F(1,943) = 104.403$, $p=2.569 \times 10^{23}$], suggesting that AN fibers followed the fine structure of the stimulus and thus represented the frequency shift in their spiking pattern (*Figure 5A–J and L*). Δf% significantly interacted with stimulus condition and BF-class [interaction: $F(2,943) = 20.292$, $p=2.353 \times 10^{9}$; $F(1,943) = 31.050$, $p=3.282 \times 10^{-8}$, respectively]. In response to 400/1600 Hz, the shifts affected the temporal representation most (i.e. higher Δf% resulted in higher TFS log-z-ratios). Additionally, TFS log-z-ratios of low-BF fibers were influenced more by the inharmonic shifts than those of high-BF fibers (again, higher Δf% results in higher TFS log-z-ratios).

TFS representation by the TFS log-z-ratio metric was not significantly different between AN fibers from different gerbil groups ($p=0.154$). Also, no interaction between group and Δf% occurred. Together, this indicates that aged fibers (*Figure 5D, H and L*) and fibers that survived the treatment with ouabain (*Figure 5C and G*) had responses that did not differ from young-adult (*Figure 5A, E and I*) and sham-treated fibers (*Figure 5B, F and J*). As expected, TFS log-z-ratios of AN fibers responding to the stimulus condition with higher $f_c$ were lower, compared to those responding to stimuli with lower $f_c$ (200/1600 Hz: 0.987, 400/1600 Hz: 1.111, 400/3200 Hz: 0.394; [main effect: $F(2,943) = 7.495$, $p=5.894 \times 10^{-4}$]). This reflects the known decrease in phase-locking with increasing stimulus frequency (*Versteegh et al., 2011*).

## Representation of f₀ was enhanced in AN fibers of old gerbils

As a metric for the relative representation of TFS vs. stimulus envelope, the ENV/TFS log-z-ratio was used (see Methods section 'Neural recordings - Data analysis'). This metric reflects the relative

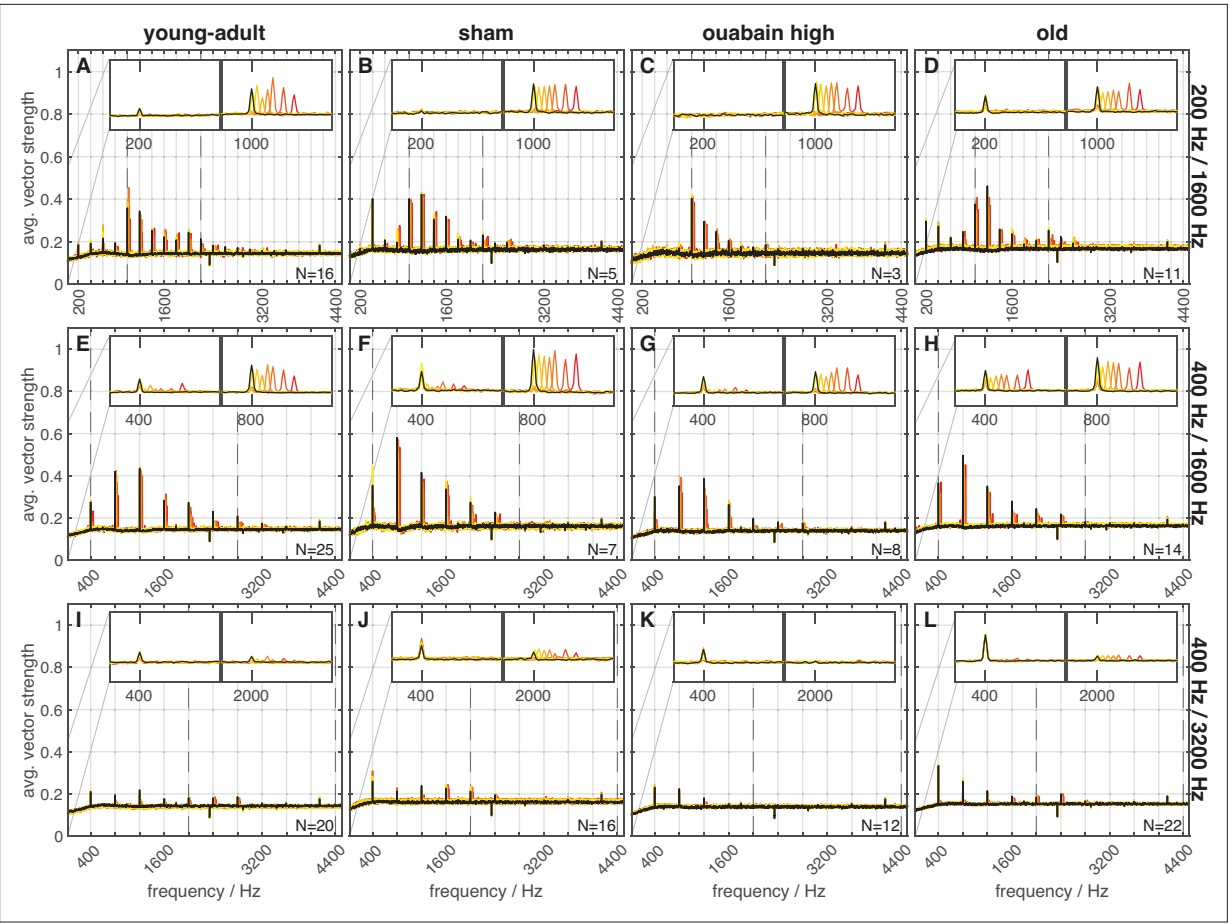

**Figure 5.** Frequency shift representation in the neural responses differed between groups and stimulus conditions. Vector strength frequency spectra of responses to harmonic and inharmonic stimuli are averaged across repetitions and fibers. Subject groups are arranged in columns and stimulus conditions in rows. Colors code the frequency shift Δf in percent of f₀, with the same color code as in **Figure 3**. Dashed lines indicate the limits of the stimulus bandpass filters. The inset panels show enlarged examples of the spectra ranging around the envelope frequency (f₀, 200 or 400 Hz) and the fine structure (center) frequency (temporal fine structure TFS peak frequency; 1000, 800, or 2000 Hz). The thin gray lines mark the y-axis scaling of the inset panels. In each panel, the maximal number of fibers (with 40 stimulus repetitions per fiber) that the spectra are based on is indicated; note that not all fibers were tested with each frequency shift Δf.

emphasis in the AN fibers' responses of either TFS (negative ENV/TFS log-z-ratio) or envelope locking (positive ENV/TFS log-z-ratio).

Here, over all animal groups, Δf% had a marginally significant effect on the ENV/TFS log-z-ratio [main effect: F(1,943) = 4.278, $p$=0.039] and stimulus condition had a highly significant effect on the ENV/TFS log-z-ratio [main effect: F(2,943) = 43.929, $p$=5.744 × 10⁻¹⁹]. Post hoc comparisons revealed that, for 400/3200 Hz, envelope representation (mean ENV/TFS log-z-ratio=0.263) was stronger than fine structure representation, indicating stronger phase-locking of AN fibers to low-frequency stimulus components. Furthermore, the stimulus condition showed a significant interaction with gerbil groups [interaction: F(6,943) = 11.439, $p$=2.242 × 10⁻¹²]. In old gerbils, the strength of phase-locking was nearly equal to the envelope (**Figure 5D**, left inset; **Figure 6D**) and to the TFS (**Figure 5D**, right inset; **Figure 6D'**), in response to 200/1600 Hz.

In response to 400/3200 Hz, fibers from old gerbils also showed better envelope locking (**Figures 5L and 6L**). Since TFS representation was unchanged in old animals (see above), this result reflects enhanced envelope coding by AN fibers obtained in old gerbils, compared to all other animal groups (**Figure 5A–D and I–L**, left insets). In contrast, AN fibers of young-adult gerbils, irrespective of treatment, tended to respond strongly to the fine structure of the stimulus, especially for conditions with lower f_c of 1600 Hz (i.e. showed VS peaks at the respective, shifted frequencies; **Figure 5A–H**, right insets; **Figure 6A'–C' and E'–G'**).

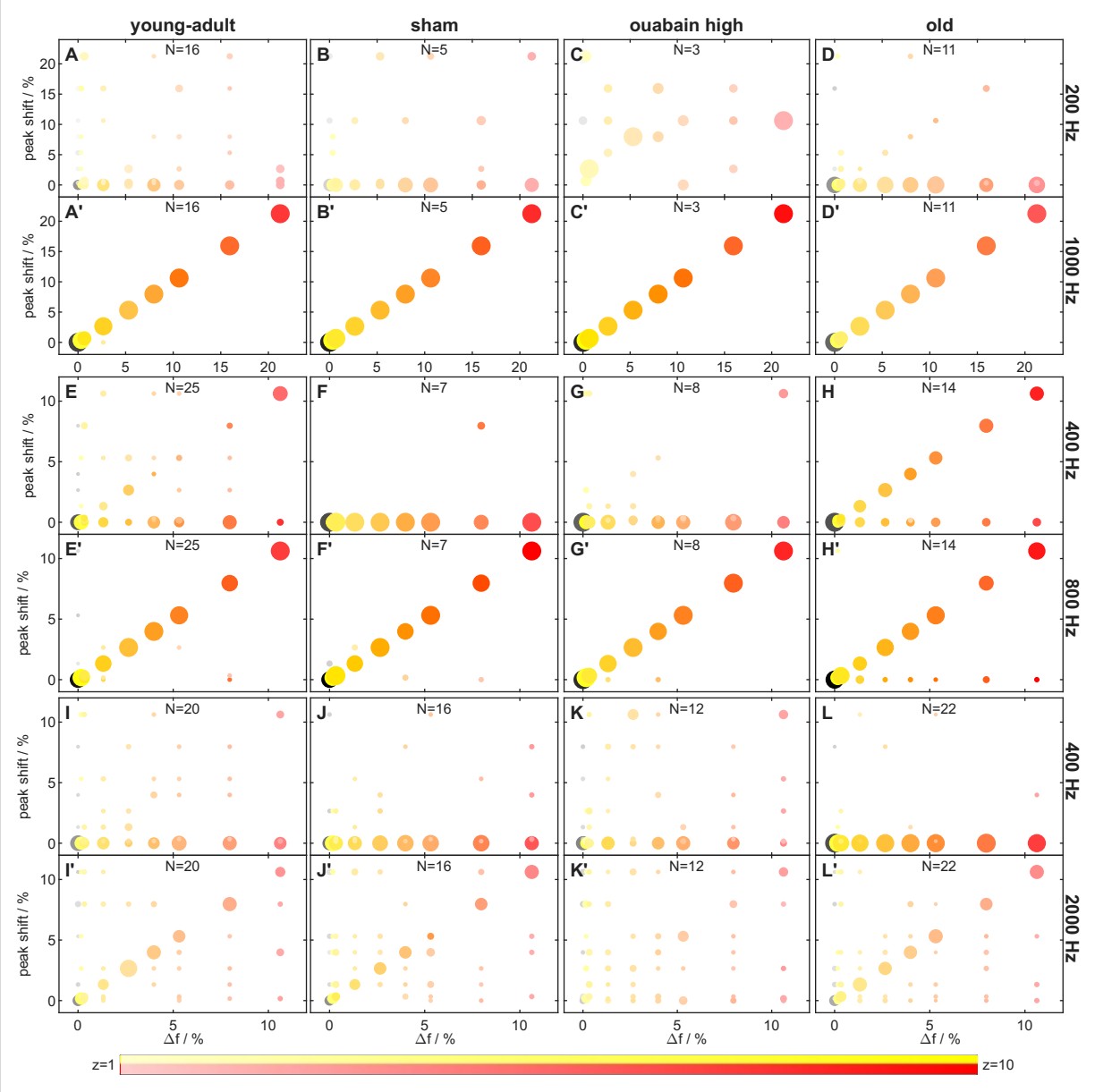

**Figure 6.** Frequency shift representation differed between envelope and temporal fine structure (TFS) frequency ranges. Frequency shift of the z-value peak versus stimulus frequency shift Δf. Subject groups are arranged in columns and stimulus conditions in double rows. Odd rows show the data at $f_0$ and even rows at $f_{maxpeak}$ (the frequency of maximal average response, *Figure 12*). Each circle represents four dimensions: the stimulus frequency shift Δf (position on the x-axis), the frequency shift of the z-value peak (position on the y-axis), the average vector strength (VS, z-score) across all respective fibers (color saturation, see legend at the bottom), and the percentage of fibers with a frequency shift of the z-value peak at the corresponding stimulus frequency shift Δf (circle radius). The largest circles correspond to 100% of fibers (e.g. in panel **C**), medium-sized circles to values around 50% (e.g. panel **J**). Colors code the frequency shift Δf in percent of $f_0$, with the same color code as in *Figure 3*. In each panel, the maximal number of fibers (with 40 stimulus repetitions per fiber) that the z-value peaks are based on is indicated; note that not all fibers were tested with each frequency shift Δf.

Over all animal groups, low-SR fibers had, on average, positive ENV/TFS log-z-ratios, that is, phase-locked more strongly to the stimulus envelope, whereas high-SR fibers showed, on average, negative ENV/TFS log-z-ratios, that is, phase-locked more strongly to the TFS [main effect: F(1,943) = 29.138, $p$=8.533 × 10$^{-8}$].

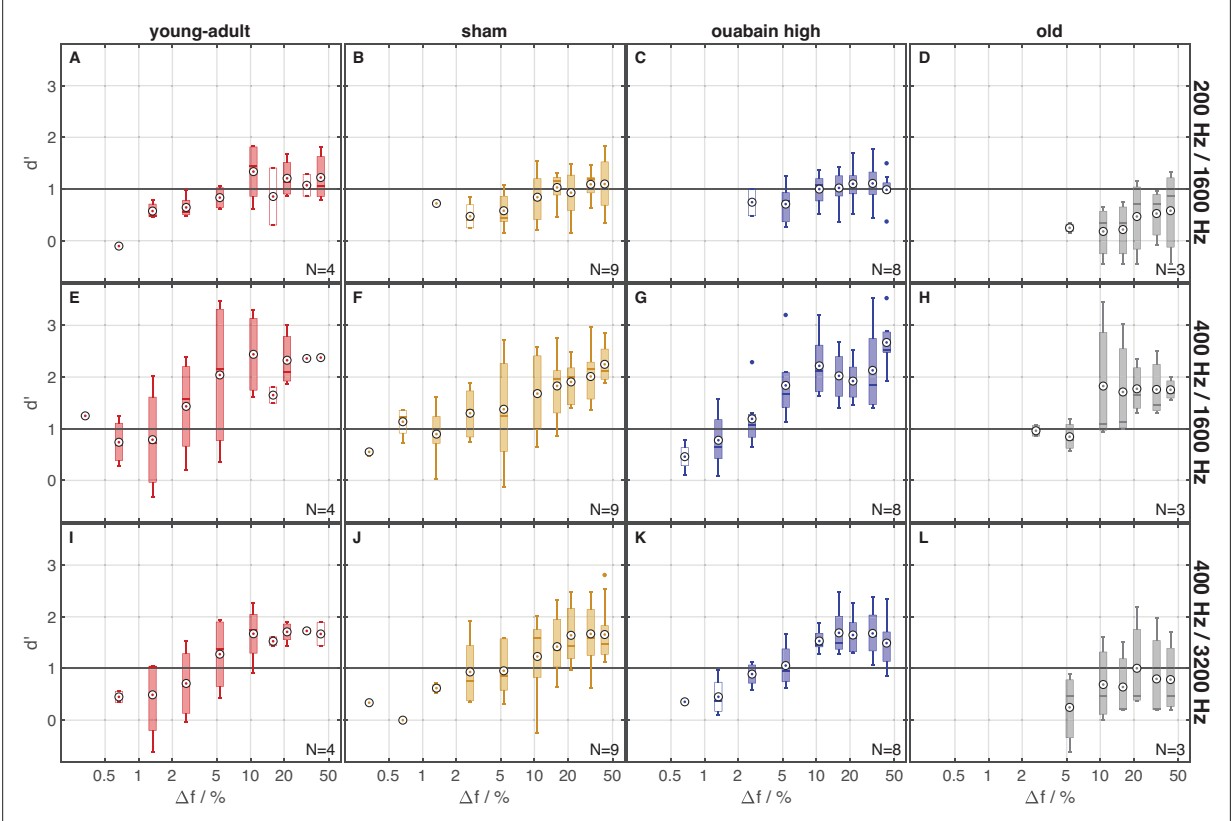

**Figure 7.** The gerbils' discrimination performance improved with larger frequency shifts. Sensitivity index d' for behavioral discrimination, as a function of stimulus frequency shift Δf in percent of $f_0$. Reference lines at d'=1 indicate the assumed threshold value for meaningful discrimination performance. Subject groups are arranged in columns and stimulus conditions in rows. Colors code the subject group. Circled black dots show the mean across subjects, boxes show the median and interquartile ranges, whiskers show the extrema, except for outliers, which are displayed as colored dots. Data points are considered outliers if they lie beyond 1.5 times the distance between median and upper or lower quartile, respectively. For each panel, the maximal number of subjects in the respective group is indicated. Unfilled boxes mark conditions completed by less than half of those subjects in the respective group.

## Age, but not synapse loss, affected behavioral discrimination of TFS1 stimuli

To investigate the effect of synapse loss on performance in the TFS1 task, the behavior of young-adult gerbils, without and with cochlear synaptopathy, and old gerbils was compared for the different stimulus conditions defined by the combinations of $f_0$ and $f_c$. The sensitivity of the four groups of gerbils in the different stimulus conditions is illustrated in *Figure 7*, in relation to the frequency shift Δf%. The sensitivity of young-adult gerbils of the different treatment groups was quite similar, whereas the old gerbils had considerably lower sensitivity. A GLMM ANOVA with the sensitivity *d'* as dependent variable, condition and treatment group as factors, and frequency shift Δf% as a covariate (only trials with Δf%>5 included, for which data from all treatment groups were obtained), revealed significant main effects of treatment group [$F_{(3,387)}$=8.18, *p*=2.725 × 10$^{-5}$], stimulus condition [$F_{(2,387)}$=22.71, *p*=4.736 × 10$^{-10}$], and Δf% [$F_{(1,387)}$=20.457, *p*=8.121 × 10$^{-6}$]. No significant interactions were observed. There was a general increase in sensitivity with increasing Δf%. Old gerbils showed lower sensitivities than most young-adult gerbils, irrespective of their treatment (all *p*<0.002, Bonferroni corrected). There was no significant difference in sensitivity between the treatment groups of young-adult gerbils. To separately investigate the effects of $f_0$ and harmonic number, *N*, on sensitivity, we conducted planned comparisons in two additional GLMM ANOVAs, each containing only the data of two conditions, with the same dependent variable and factors as the initial ANOVA. The planned comparison with the data of stimulus conditions 200/1600 Hz and 400/1600 Hz was used to investigate the effect of $f_0$ (which inherently includes a change in harmonic number, N) on sensitivity. This

ANOVA revealed an effect of $f_0$, with a higher sensitivity for increasing $f_0$ [F(1,257)=45.001, $p$=1.248 × $10^{-10}$]. Furthermore, an effect of $\Delta$f% [F(1,257)=13.539, $p$=2.847 × $10^{-4}$] was observed. Similar to the previous analysis, the main effect of gerbil group [F(3,257)=4.878, $p$=2.570 × $10^{-3}$] reflected the low sensitivity of the old gerbils compared to all other groups. The second planned comparison with the data of stimulus conditions 400/3200 Hz and 400/1600 Hz, with equal $f_0$, investigated the dependence of sensitivity on $f_c$. An effect of $f_c$ [F(1,256)=12.656, $p$=4.459 × $10^{-4}$] was shown, with decreasing sensitivity for increasing $f_c$. As in the previous comparison, this ANOVA also revealed increased sensitivity with increasing $\Delta$f% [F(1,256)=13.484, $p$=2.929 × $10^{-4}$], whereas the main effect of gerbil group [F(3,256)=4.692, $p$=3.299 × $10^{-3}$] again reflected the lower sensitivity of the old group compared to all other groups. No interactions were observed in either planned comparison.

The frequency-shift threshold reflects $\Delta$f% for a criterion of $d'$=1 and was calculated for each gerbil and condition. If no threshold could be derived because all $d'$ values were below 1, a threshold of 100% $f_0$ was assumed for further statistical analysis. On average, old gerbils were only able to discriminate inharmonically shifted TFS1 stimuli for the condition 400/1600 Hz (see $d'$ values >1 in *Figure 7D, H and L*). All young-adult gerbil treatment groups achieved average sensitivities above threshold for all stimulus conditions.

Frequency shift thresholds for the three different stimulus conditions are shown in *Figure 8*. A GLMM ANOVA was conducted with $\Delta$f% thresholds as the dependent variable and condition and treatment group as factors. There were main effects for stimulus condition [F(2,60)=8.99, $p$<0.0005] and treatment group [F(3,60)=6.84, $p$<0.0005], with no interaction between both factors. Average thresholds were not significantly different between the 200/1600 Hz and 400/3200 Hz conditions, while thresholds were lower in the 400/1600 Hz than in the 200/1600 Hz condition ($p$<0.0005). The pairwise comparison revealed that old gerbils had significantly higher thresholds than gerbils of all other animal groups (all $p$<0.005).

No differences were observed between the young-adult gerbil groups with different treatments. To evaluate the effect of $f_0$ on the threshold, a planned comparison with the data of stimulus conditions 200/1600 Hz and 400/1600 Hz (i.e. harmonic numbers 8 and 4, respectively) was conducted in an additional ANOVA, with frequency-shift thresholds as dependent variable, and condition and gerbil group as factors. A main effect of condition was revealed, indicating that $f_0$ (respectively harmonic number N at 1600 Hz $f_c$) did affect the threshold [F(1,40)=15.219, $p$=3.577 × $10^{-4}$]. There was no significant effect of gerbil group; all groups showed similarly low thresholds for the 400/1600 Hz condition. A second additional ANOVA with the data of stimulus conditions 400/3200 Hz and 400/1600 Hz (i.e. harmonic numbers 8 and 4, respectively) investigated the dependency of threshold on $f_c$ (respectively harmonic number N). An effect of gerbil group [F(3,40)=10.460, $p$=3.272 × $10^{-5}$] and condition [F(1,40)=16.213, $p$=2.455 × $10^{-4}$] was observed. The pairwise comparison revealed lower thresholds with decreasing $f_c$ (respectively harmonic number N) and a significant difference between untreated and old gerbils (all $p$≤0.001), whereas the young-adult gerbil treatment groups did not differ.

It has been suggested that human subjects compensate difficulties in speech perception in a noisy background due to deficits in TFS processing by increasing their listening effort (*Parthasarathy et al., 2020*). We investigated the possibility that the ouabain-treated young-adult animals may have compensated for their loss of synapses by increasing their listening effort, predicted to result in longer response latencies. To this end, we analyzed the distributions of response latencies for a range of frequency shifts between 2.65% and 31.88% applying a mixed model ANOVA with response latency as the dependent variable and group (ouabain-treated young-adult animals vs. untreated young-adult animals), stimulus conditions (200/1600, 400/1600, and 400/3200), and frequency shift as factors. Response latency was significantly affected by the extent of the frequency shift [F(4,3451)=7.516, $p$=5.027 × $10^{-6}$] and by the stimulus condition [F(2,3451)=3.621, $p$=2.687 × $10^{-2}$], but not by the group [F(1,3451)=15.219, $p$=3.719 × $10^{-1}$]. There were no significant interactions. Thus, we conclude that there is no evidence that ouabain-treated young-adult animals compensated for a putative perceptual deficit by increasing the listening effort.

In contrast, a similar analysis comparing untreated young-adult and old animals revealed significantly longer response times in old animals compared to young-adult animals (difference 66 ms, [F(1,2251)=13.156, $p$=2.930 × $10^{-4}$]). Despite the longer response latencies in old animals, suggesting increased listening effort, the performance of the old animals did not reach that of young-adult animals.

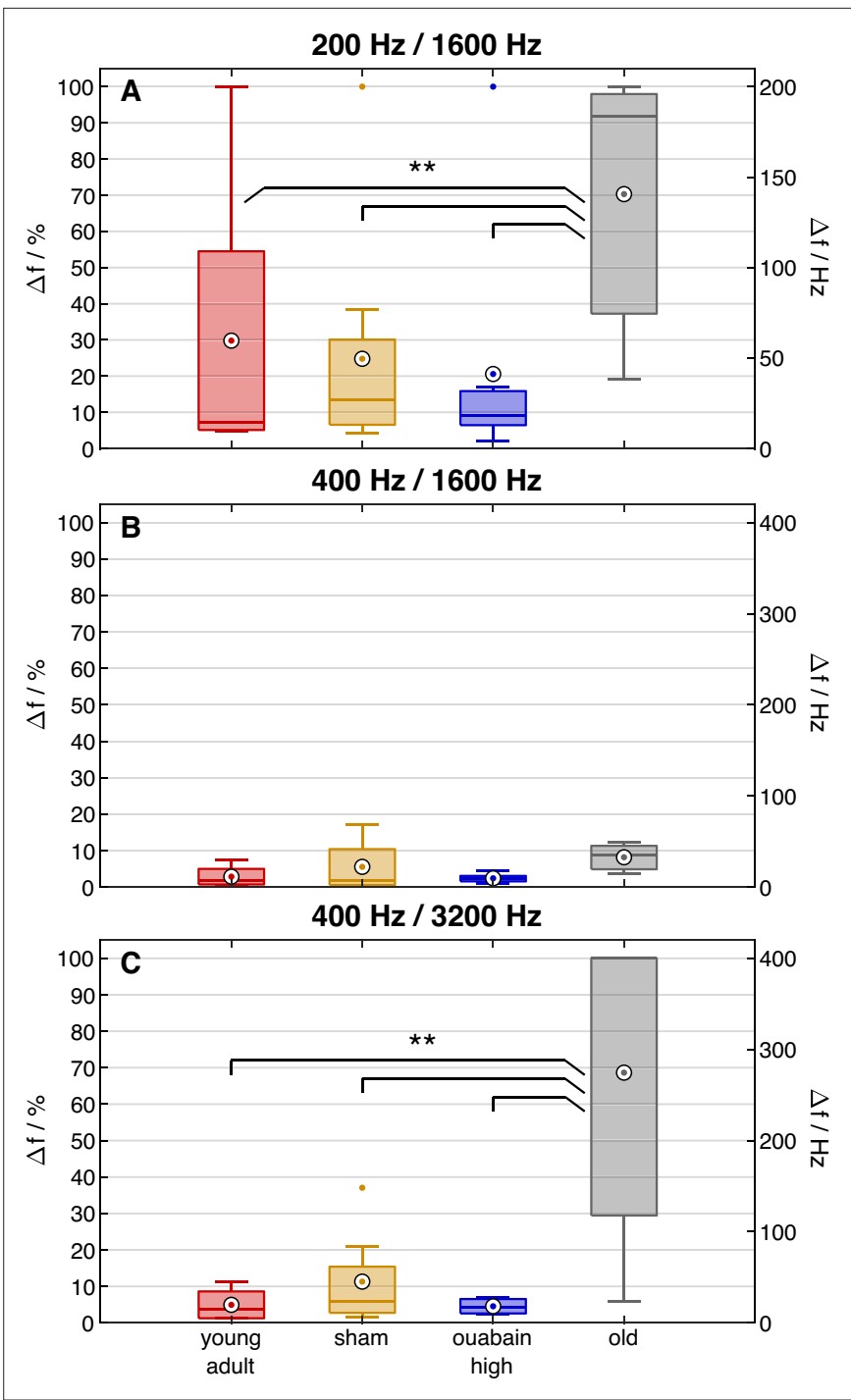

**Figure 8.** Old gerbils were typically unable to perceive frequency shifts, whereas ouabain treatment did not impair discrimination. Behavioral discrimination thresholds based on the data shown in **Figure 7**. Thresholds were defined as the lowest (linearly interpolated) stimulus frequency shift Δf in percent of $f_0$ at which the sensitivity index d' crossed d'=1. The threshold was set to 100% if this criterion was never met. Colors code the subject group. Circled dots show the mean across subjects, boxes show the median and interquartile ranges, whiskers show the extrema, except for outliers, which are displayed as colored dots. Data points are considered outliers if they lie beyond 1.5 times the distance between median and upper or lower quartile, respectively. The y-axes at the right show the same frequency shifts in Hz. Significance levels are **p<0.002 for all three comparisons, young-adult vs. old $p$=1.53 × $10^{-3}$, sham vs. old $p$=1.11 × $10^{-3}$, ouabain high vs. old $p$=0.382 × $10^{-3}$.

In summary, the behavioral data showed that the old gerbils differed from all other treatment groups of young-adult gerbils. This result suggests that age considerably affected both perceptual sensitivity and thresholds, while synapse numbers had little effect.

## Discussion

The ability to process temporal fine structure (TFS) of sounds is a key factor for speech perception. Its age-related decline has been held responsible for compromised speech comprehension in the elderly, especially in challenging noisy acoustic backgrounds (*Füllgrabe et al., 2014*; *Moore, 2014*; *Moore, 2019*). It has been suggested that age-related or noise-induced synaptopathy results in the observed perceptual deficits regarding the TFS of sounds (e.g. *Moore, 2014*; *Moore, 2019*). However, so far, a direct test of this hypothesis is lacking. Here, we present data that investigate the performance of the Mongolian gerbil in the TFS1 test (*Moore and Sek, 2009*), a psychoacoustic paradigm that has been applied in human subjects to evaluate the ability to discriminate between stimuli based on the TFS. We determined the TFS1 test performance in young-adult gerbils and compared this to the performance of old gerbils (≥36 months of age, corresponding to humans ≥60 years of age, *Castaño-González et al., 2024*), demonstrating an age-related decline in TFS sensitivity similar to that found in human subjects (*Füllgrabe et al., 2014*; *Moore et al., 2012*). However, young-adult gerbils with experimentally induced synaptopathy (by applying ouabain), to a degree corresponding to the synaptopathy in old gerbils (*Steenken et al., 2021*, this study), showed no decline in behavioral TFS sensitivity. This result suggests that reduced synapse numbers alone cannot explain the perceptual deficits for TFS in old subjects. Auditory-nerve (AN) fiber responses of young-adult, young-adult ouabain-treated, and old gerbils, elicited by the stimuli presented in the TFS1 test, indicate that the representation of TFS by AN fibers is not affected by ouabain treatment or age. In old gerbils that show compromised TFS1 test performance behaviorally, however, the representation of the signal envelope by AN fibers was enhanced compared to the other groups. We propose that this enhancement of the signal envelope cue, which does not differentiate the stimuli in the TFS1 test, may cause the perceptual deficits in old gerbils by overriding the usability of the TFS cues. Thus, cue representation may be associated with the perceptual deficits, but not reduced synapse numbers, as originally proposed.

### Gerbil as a model for human temporal-fine-structure perception

In the TFS1 test, subjects must discriminate an inharmonic (I) complex from a harmonic (H) complex, in which the frequency difference between the components determines the period of the envelope. The I complex is created by shifting all components of the H complex by a fixed frequency. This shift results in pairs of I and H complexes with similar envelopes but different TFS. Thus, TFS is the only acoustic cue which allows to discriminate the stimuli. Excitation-pattern differences due to the shift of the frequency components in the I complex compared to the H complex can be eliminated in the TFS1 test by appropriate spectral filtering (e.g. *Marmel et al., 2015*). In addition, the gerbil has larger auditory filter bandwidths than human subjects (*Kittel et al., 2002*). Since we used a similar filter bandwidth for the gerbil as that applied in studies involving humans (e.g. *Eipert et al., 2019*; *Marmel et al., 2015*; *Moore and Sek, 2009*), cochlear excitation pattern differences are unlikely to have provided usable cues for discrimination by the gerbils in the present study, at least for conditions with a harmonic number N=8 (200/1600 Hz, 400/3200 Hz). However, it appears possible that for the stimulus condition 400/1600 Hz (harmonic number N=4), the gerbils' discrimination may have relied on excitation pattern differences (see model results in *Figure 9*). The modeled cochlear excitation patterns also suggest that the noise masker in which the I and H complexes were presented ensured that the stimuli did not produce usable distortion products for the gerbils. Thus, as in the study with human subjects, the only usable cue for the discrimination for conditions with a harmonic number N=8 was the TFS.

*Figure 10* provides an overview of gerbil TFS1 thresholds in comparison to human thresholds. Human thresholds were selected from studies involving stimuli with spectral components and fundamental frequencies ($f_0$) similar to those presented to the gerbils in the present study, allowing a comparison of gerbil and human performance. There is a broad overlap between gerbil and human TFS1 thresholds. Both in humans and gerbils, old subjects showed higher (i.e. worse) TFS1 thresholds than young subjects (for human studies see *Eipert et al., 2019*; *Füllgrabe et al., 2014*; *Moore*

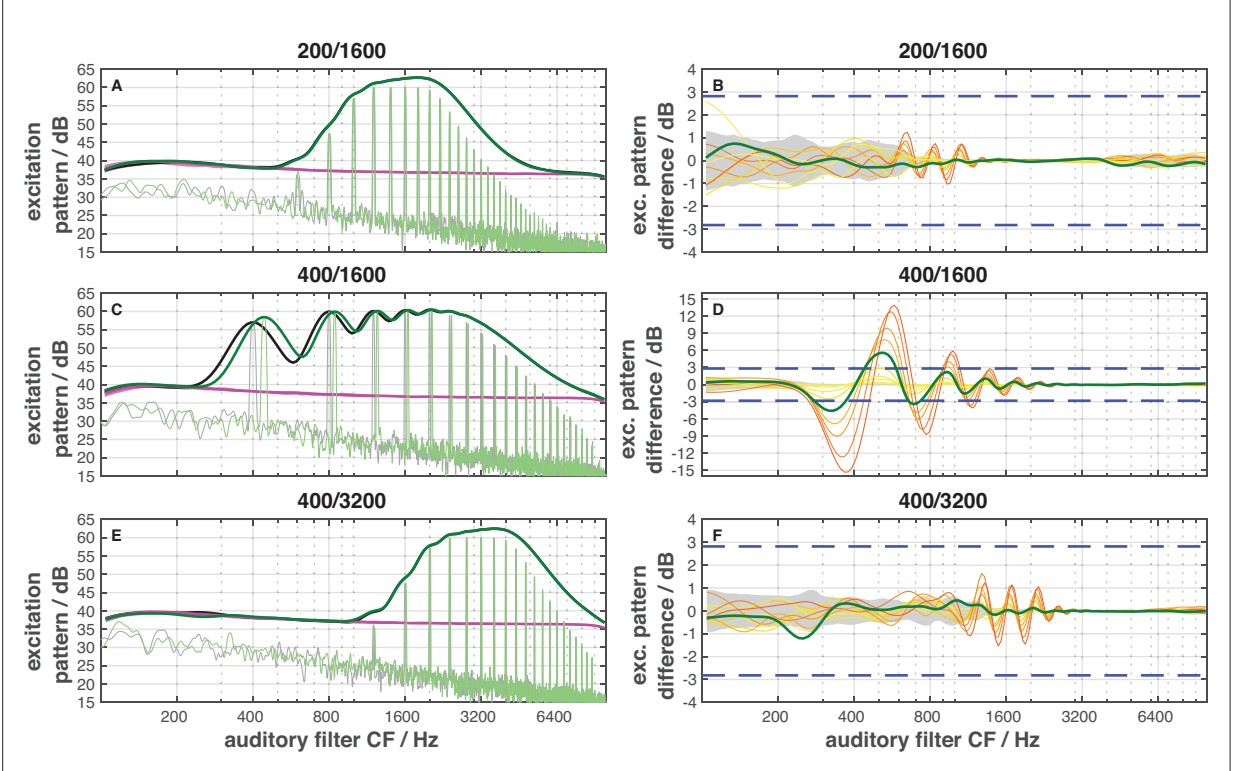

**Figure 9.** Gerbil excitation patterns for harmonic (H, thick black lines in panels **A**, **C**, and **E**) and inharmonic (I) stimuli close to the frequency shift at the TFS1 threshold (thick dark green lines in panels **A**, **C**, and **E**) and their differences (panels **B**, **D**, and **F**) for the three stimulus conditions. Thin grey and green lines in panels **A**, **C**, and **E** show the physical sound spectra for the corresponding harmonic (H) and inharmonic (I) stimuli, respectively. Red lines in panels **A**, **C**, and **E** show the excitation pattern elicited by the pink noise masker. Since the level of excitation produced by the pink noise is less than 30 dB below that produced by the complex tones, distortion products will be masked. Excitation patterns were calculated with the assumption that the gerbil auditory filter bandwidth is 1.8 times the human auditory filter bandwidth (see **Kittel et al., 2002**). The thick green lines in panels **B**, **D**, and **F** show the differences between the excitation pattern elicited by the harmonic (H) reference and the inharmonic (I) stimuli close to the frequency shift at the TFS1 threshold. The thin yellow and orange lines show these differences for the different frequency shifts of the inharmonic (I) stimuli (color codes as in **Figure 3**). The gray-shaded areas in panels **B**, **D**, and **F** indicate the 5%/95% percentiles of the amplitude statistics in the harmonic (H) reference signals. The blue dashed lines in panels **B**, **D**, and **F** show the gerbil threshold for detecting a change in the intensity of a 70 dB 1 kHz tone (**Sinnott et al., 1992**). Only for the condition 400/1600 Hz does the change in the level of the excitation pattern due to the frequency shift exceed the threshold for detecting the intensity difference. For this condition, the gerbils could solve the task, at their TFS1 threshold, by comparing the difference in excitation elicited by the harmonic (H) and frequency-shifted inharmonic (I) stimulus. For the other conditions, this change is less than the intensity difference limen. Thus, for these two conditions with harmonic number N of 8, the gerbils cannot rely on differences in the excitation patterns but must solve the task by comparing the temporal fine structure.

et al., 2012). This similarity indicates that the gerbil is a good model for investigating the mechanisms underlying TFS perception that may also apply to humans. Behavioral TFS1 thresholds in young-adult gerbils with ouabain-induced synaptopathy, similar to that typical for old gerbils, were similar to TFS1 thresholds in untreated young-adult gerbils. This result suggests that increased behavioral thresholds in old subjects were not due to the reduced number of synapses. Thus, in the gerbil, the hypothesis that age-related deterioration of TFS perception is a direct consequence of synaptopathy can be rejected.

The observed extent of age-related or noise-induced loss of type-I afferent synapses on IHC varies widely between species and studies. For example, in ageing CBA/CaJ mice, mean losses of between 20 and 50% of afferent synapses (depending on cochlear location and precise age) were reported (**Sergeyenko et al., 2013**; **Kobrina et al., 2020**). Humans showed more pronounced losses of peripheral axons, of 40–100%, again depending on cochlear location, precise age, and noise history (**Wu et al., 2019**; **Wu et al., 2021**). The age-related and induced synapse losses in our gerbils were in a more moderate range, around 20% (**Steenken et al., 2021**, this study). Thus, it is possible that a more severe, induced synaptopathy would have resulted in behavioral deficits in young-adult gerbils.

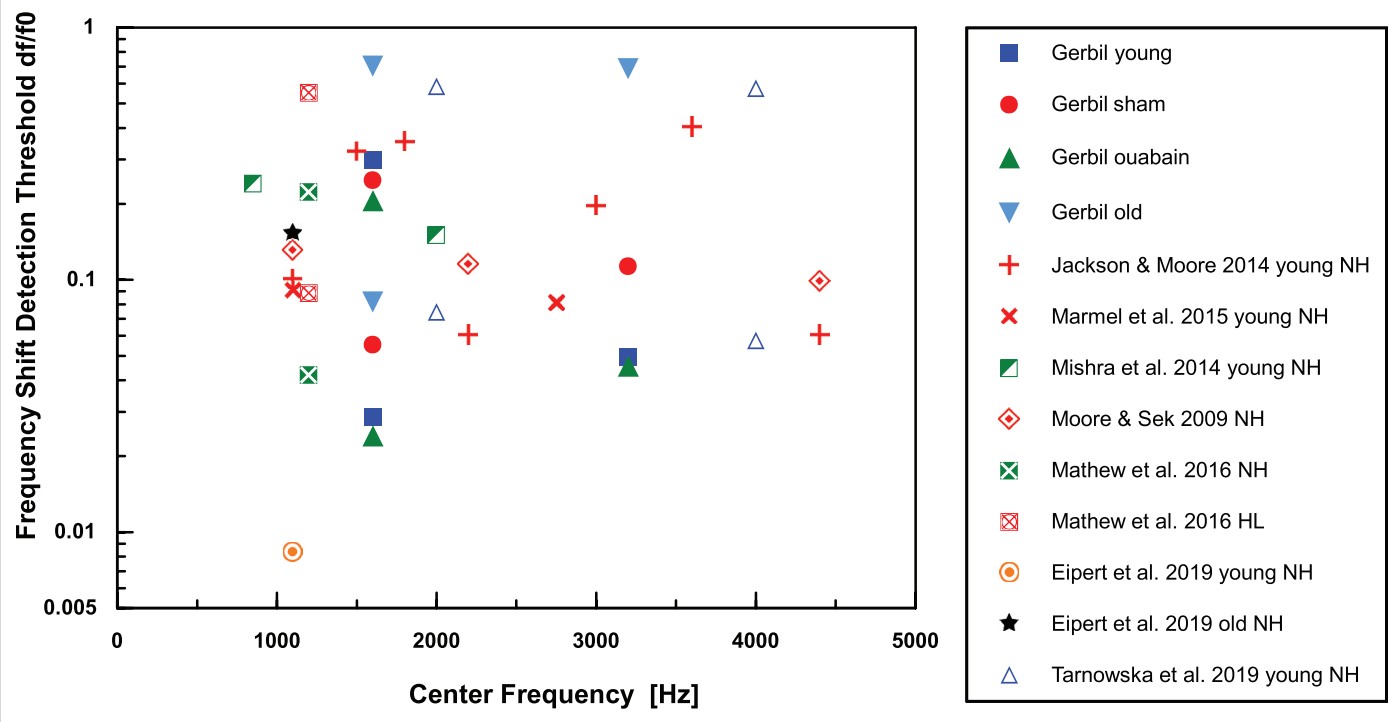

**Figure 10.** Frequency shift detection thresholds of gerbils and humans for TFS1 test stimuli in relation to the center frequency $f_c$ of the harmonic complex. Filled symbols represent gerbil data from the present study. Human data are from published material with the limitation that the $f_c$ was in the range from 500 Hz to 4500 Hz and the fundamental $f_0$ of the harmonic complex was in the range of 100 Hz to 400 Hz, to make the range of parameter values similar to those of the present study (for references see legend, normal hearing (NH), hearing loss (HL)). The higher and lower thresholds shown for the gerbil data reflect thresholds at $f_c$ of 1600 Hz for fundamentals $f_0$ of 200 Hz and 400 Hz, respectively.

However, in the absence of additional noise or pharmacologically induced damage, our study provides strong evidence for other factors causing temporal processing problems with advancing age. Our 3-year-old gerbils are approximately comparable to a 60-year-old human (*Castaño-González et al., 2024*) with beginning but not yet clinically relevant hearing loss (*Hamann et al., 2002*).

### Auditory-nerve fiber responses do not explain differences in behavioral sensitivity

Since the use of excitation patterns by the gerbil is unlikely due to its larger auditory-filter bandwidths for stimulus conditions of 200/1600 Hz and 400/3200 Hz (*Figure 9*), we explored the temporal responses of AN fibers for alternative explanations of compromised TFS perception. TFS representation requires the ability of AN fibers to phase-lock (reviewed in *Joris et al., 2004*; *Rose et al., 1967*). Here, we showed that, throughout all tested gerbil groups, the H and I complexes were neurally distinguishable based on phase-locking to the stimulus fine structure (*Figures 5 and 6*). In contrast, no distinction was possible based on the average firing rate of AN fibers (*Figure 3*). Neural phase-locking reached highest z-values for stimuli with $f_c$ of 1600 Hz and were lower for 3200 Hz (*Figure 6*), which reflects the known decrease in phase-locking of AN fibers for stimuli above 1.5 kHz (*Versteegh et al., 2011*). Several studies in animal models of cochlear dysfunction showed that phase locking is not compromised in single AN fibers, including quiet-aged gerbils (*Heeringa et al., 2020*), noise-exposed cats (*Miller et al., 1997*), chinchillas (*Henry and Heinz, 2012*; *Kale et al., 2013*), and kanamycin-treated guinea pigs (*Harrison and Evans, 1979*). Consistent with these findings, the surviving AN fibers in old gerbils did not show compromised TFS representation of I and H complexes compared to young-adult gerbils (*Figures 5 and 6*).

Old gerbils showed significant synapse loss compared to untreated young-adult gerbils at the cochlear location corresponding to 2 kHz (*Figure 1* and *Table 1* for all frequencies), consistent with what was shown previously (*Steenken et al., 2021*). Ouabain, a cardiac glycoside that inhibits $Na^+$-$K^+$-ATPase pumps of AN fibers (*O'Brien et al., 1994*), applied to the round window, results in a

dose-dependent loss of AN fibers in gerbils (*Bourien et al., 2014*; *Lang et al., 2005*; *Schmiedt et al., 2002*). In our hands, a low dose (40 µM) of ouabain had no effect at the cochlear location equivalent to 2 kHz. In contrast, a higher dose of 70 µM resulted in a loss that was similar to that observed in old gerbils (*Figure 1*). The surviving fibers of ouabain-treated gerbils showed no deterioration of frequency tuning and TFS representation compared to untreated or sham-treated young-adult gerbils (*Figures 2 and 5A, C, E and G*); however, the behavioral outcomes of this study clearly showed that only old gerbils had deteriorated discrimination abilities between H and I complexes (*Figure 8*). We thus showed conclusively that neither synapse loss, to an extent typical for old gerbils, nor impaired TFS representation, at the level of single AN fibers, could explain the deteriorated perception in old gerbils.

## Enhancement of confounding/distracting AN cues and changed central processing may explain the behavioral deficit in old subjects

Surviving AN fibers in old gerbils did not show compromised fine-structure representation (*Figures 5 and 6*). However, old gerbils exhibited deteriorated performance in the behavioral test in response to TFS1 test stimuli (*Figures 7 and 8*). Despite the overall robust phase-locking ability, other pathologies are known to develop with advancing age.

First, the age-related atrophy of the stria vascularis within the lateral wall of cochleae (*Gratton and Schulte, 1995*) results in a decreased endocochlear potential (*Schulte and Schmiedt, 1992*), which in turn increases the threshold (i.e. decreases the sensitivity) of AN fibers (*Schmiedt et al., 1996*). Envelope representation, measured as synchronization to amplitude-modulated stimuli, is non-monotonic with sound level, with a pronounced peak near the threshold of the fiber (*Heeringa et al., 2023*; *Joris and Yin, 1992*; *Kale and Heinz, 2010*). Thus, the typical sensitivity loss in old subjects will cause a difference in neural representations between young-adult and old subjects, even if the test stimulus levels are kept constant. Alternatively, stimulus levels can be raised for old subjects to an equivalent sensation level that compensates for their sensitivity loss. Consistent with that prediction, AN fibers of old gerbils show enhanced representation of the fundamental frequency $f_0$ of speech vowels, compared to fibers from young adults, when responding to fixed-level stimuli (*Heeringa et al., 2023*), but do not show enhanced envelope representation in response to frozen noise presented at similar sensation level (*Heeringa et al., 2020*). AN fibers of old gerbils of the same age as in the present study invariably show threshold loss, although to variable extents (*Hellstrom and Schmiedt, 1990*; *Steenken et al., 2021*, this study). Thus, since TFS1 stimuli in the current study were presented at a fixed absolute level of 68 dB SPL, fibers of old animals were stimulated closer to their individual thresholds than fibers of young adults, which is expected to increase their phase locking to $f_0$, as observed. Our findings match those of *Kale and Heinz, 2010* for AN fibers of chinchillas with noise-induced hearing loss. After noise exposure, envelope coding was enhanced, in approximate proportion to the fiber's sensitivity loss, while the representation of TFS remained unchanged (*Kale and Heinz, 2010*). Together, these AN data suggest that both age-related and noise-induced hearing loss result in a relative deficit of TFS representation that may ultimately contribute to perceptual problems.

A decreased endocochlear potential may also affect the AN fibers' frequency selectivity, indirectly via compromised OHC function (*Ruggero and Rich, 1991*). In chinchillas with noise-induced hearing loss, deteriorated frequency tuning of AN fibers and tail hypersensitivity led to abnormal temporal coding, such that a mismatch emerged between the fibers' characteristic frequency and the dominant frequency represented in their TFS responses (*Henry et al., 2016*), including responses to TFS1 stimuli (*Kale et al., 2013*). However, this is an additional effect, unique to noise damage, and not likely to apply to our aged gerbils. In old gerbils, the frequency selectivity of the AN fibers' sensitive tuning-curve tips is not significantly affected (*Heeringa et al., 2020*; *Hellstrom and Schmiedt, 1996*, this study) and there is no evidence for tail hypersensitivity (*Hellstrom and Schmiedt, 1996*), consistent with only mild OHC pathologies in aging gerbils (*Tarnowski et al., 1991*; *Adams and Schulte, 1997*; *Steenken et al., 2024*). Furthermore, in chinchillas treated with furosemide, a drug that reversibly reduces the endocochlear potential, the frequency tuning curves of AN fibers were affected in their tip-to-tail ratios; however, much less drastically than after noise damage (*Henry et al., 2019*). Together, this suggests that AN fibers in aged animals, without confounding noise-induced hearing loss, experience only mild impairments of frequency tuning that did not cause the changes to temporal coding observed in our old gerbils.

Second, and moving beyond cochlear pathologies, age-related changes in GABAergic processing in the central auditory system do occur (*Caspary et al., 1995*). Such changes were also specifically shown for the gerbil (*Gleich et al., 2014*; *Kessler et al., 2020*) and were shown to affect temporal processing (*Gleich et al., 2003*). The gerbils used by *Kessler et al., 2020* were derived from the same population as the gerbils used for the present study and had the same age range for old animals. Therefore, it can be assumed that the gerbils in the current study showed a change in GABAergic processing in the brain that could affect processing of such supra-threshold stimuli as being used in the TFS1 test.

Third, olivocochlear efferent activity is a potential modulator of OHC gain (by medial olivocochlear neurons, MOC) and afferent activity (by lateral olivocochlear neurons, LOC). Beyond this general observation, it is, however, difficult to speculate about its specific role in the TFS1 test, as almost nothing is known about efferent activity under naturalistic conditions in a behaving animal (reviewed by *Lauer et al., 2022*). We note that efferent activity is believed to be reduced under general anesthesia (reviewed by *Guinan, 2011*) and possibly abnormal in other ways, considering the potential top-down inputs to the efferent neurons from extensive brain networks (reviewed by *Schofield, 2011*; *Romero and Trussell, 2022*). Thus, it is reasonable to assume a reduced efferent influence in our AN data, compared to the behavioral test situation. As for changes with aging, we tentatively assume more comparable efferent influences in young-adult and old gerbils. It was recently shown that, despite age-related losses in both MOC and LOC cochlear innervation, this basically reflected the loss of efferent target structures (OHC and type-I afferents), with the surviving cochlear circuitry remaining largely normal (*Steenken et al., 2024*). The main difference was an increased proportion of OHCs without any efferent innervation, predominantly in low-frequency cochlear regions (*Steenken et al., 2024*). Such OHCs are thus not under efferent control, and they are more numerous (about 10–30%) in old gerbils.

## Conclusion

We demonstrated that the gerbils' ability to behaviorally discriminate TFS1 stimuli is neither affected by synapse loss nor by alterations to TFS representation by their AN fibers. However, the elevated neural response thresholds in old gerbils enhanced the neural envelope representation. This enhancement may have overridden the still intact TFS representation.

## Materials and methods
### Experimental model and subject details

Sixty adult, agouti-colored Mongolian gerbils (*Meriones unguiculatus*) of both sexes (32 M, 28 F) were studied (*Table 2*). The gerbils originated from Charles River Laboratories and were bred in an in-house facility of the University of Oldenburg. Groups of gerbils were tested, differing in age and ouabain treatment: untreated young-adult, surgery-only young-adult, ouabain-treated young-adult, and old gerbils. Young-adult gerbils were between 3 and 19 months of age, and old gerbils were 36 months or older. All animals were housed individually or in pairs in Type IV cages, which were provided with nesting material. During behavioral testing, access to water was not limited, but animals were food-deprived, such that the weight of young-adult and old gerbils was 65–75 g and 70–85 g, respectively, to motivate the animals to perform for a food reward. All protocols and procedures were approved by the Niedersächsisches Landesamt für Verbraucherschutz und Lebensmittelsicherheit (LAVES), Germany, permit AZ 33.19-42502-04-15/1990. All procedures were performed in compliance with the NIH Guide on Methods and Welfare Consideration in Behavioral Research with Animals (*National Institute of Mental Health, 2002*).

Young-adult gerbils were on average 9.9±5.4 months of age (N=21) and old gerbils were 38.8±2.5 months old (N=17) when electrophysiological recordings were conducted and cochleae were harvested. Treated young-adult gerbils were either 15.1±3.6 (treated with a low concentration of ouabain, N=8) or 12.7±4.2 (high concentration of ouabain, N=10). All surgery-only gerbils were 17.5±1 months old (N=4). Gerbil groups had significantly different ages (univariate ANOVA [$F_{(4,55)}$=129.449; $p$=2.677 × $10^{-26}$]). Bonferroni-corrected post-hoc tests revealed that old gerbils were significantly different from all other young-adult gerbil groups with respect to age, while all young-adult gerbil groups, irrespective of treatment, did not differ.

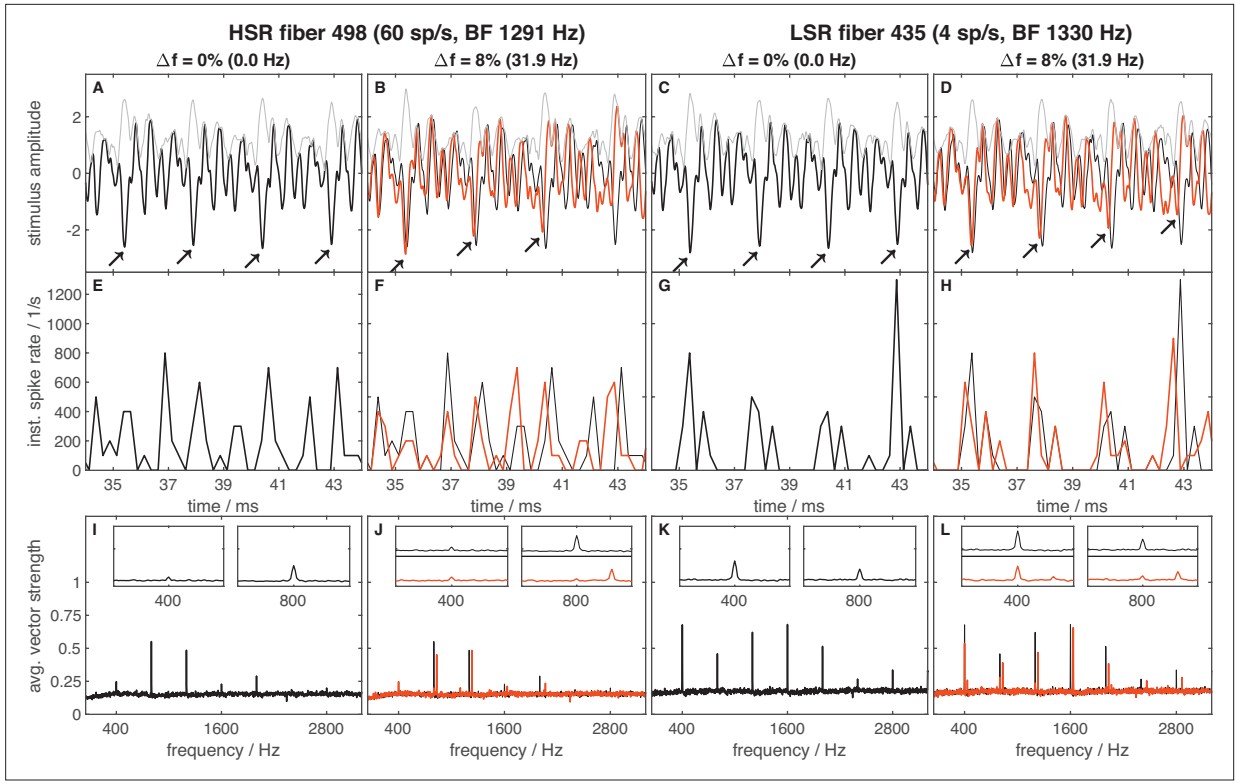

**Figure 11.** Examples illustrating stimulus waveforms and typical auditory-nerve (AN) responses. Stimulus waveforms (**A–D**), peri-stimulus time histograms PSTHs, (**E–H**), and average vector strength (VS) as a function of frequency (**I–L**). Columns 1 (**A, E, and I**) and 2 (**B, F, and J**) show data from a high (**H**) spontaneous rate (SR) fiber, columns 3 (**C, G, and K**) and 4 (**D, H, and L**) show data from a low (**L**) SR fiber. Both fibers are from a young-adult, untreated animal and are matched in best frequency (BF). The stimulus condition was $f_0$=400 Hz and $f_c$ = 1600 Hz. A and C show examples of the harmonic (H) stimulus (Δf=0%, black) and its envelope (grey), B and **D** show examples of a strongly inharmonic (I) stimulus (Δf=8%, red), its envelope (grey), and the harmonic stimulus (H, black) as a reference. The corresponding neural responses are displayed in the same colors in the second and third row, respectively. Arrow markers are added (**A–D**) to highlight the differences in fine structure between harmonic (H) and inharmonic (I) stimuli. The PSTHs (**E–H**) show the instantaneous spike rate, averaged across 40 repetitions of identical stimuli. Vector strength (VS, **I** to **L**) is shown as a function of frequency, averaged across 40 stimulus presentations. The inset panels show enlarged examples of the spectra in the envelope frequency range (400 Hz) and the fine structure frequency range (800 Hz).

## Methods details

### Stimulus paradigm

The stimuli were similar to those used for the TFS1 test in humans (for a detailed description, see *Moore and Sek, 2009*). In short, the reference stimulus was a harmonic tone (H) complex. The test stimulus that was discriminated from the reference by the gerbils in the behavioral tests or by analyses of AN-fiber responses was an inharmonic tone (I) complex created by shifting all frequency components upwards. This procedure creates H and I complexes with the same envelope periodicity. In the following, the frequency shift, Δf, will be expressed by the percentage of the fundamental frequency, $f_0$ (Weber fraction).

The $f_0$ was either 200 or 400 Hz. H and I complexes were generated with components from $f_0$ up to 9 kHz. H and I complexes were passed through the same third- or fifth-order Butterworth infinite-impulse-response band-pass filter (for AN recording and behavioral experiments, respectively), with a center frequency, $f_c$, of 1600 or 3200 Hz, to reduce possible cues related to differences in the excitation pattern on the basilar membrane (*Moore and Moore, 2003*). The 3 dB down points of the filter were separated by 7 $f_0$, which resulted in a flat spectral region of 5 $f_0$. The amplitude at the filter skirts decreased by at least 30 dB per octave (except for the third-order filter with $f_c$ of 1600 Hz and $f_0$ of 400 Hz, for which the slope was 18 dB/oct).

The phase of each harmonic was randomly chosen for each test trial, but was matched for the H and I complexes within a trial to ensure similar envelopes for both stimuli (see *Figure 11*). A pink noise

with frequencies between 100 and 11,050 Hz was added to the signal to mask distortion products possibly emerging in the inner ear and signal components outside the filter passband. The masker power spectral density was 13 dB SPL at 1 kHz, equivalent to 37 dB SPL within the gerbil's auditory equivalent rectangular bandwidth centered on 1 kHz, to mask distortion products of H and I stimuli in the gerbil (*Faulstich and Kössl, 1999*; *Bourien et al., 2014*). The level of each component in the tone complex was 60 dB SPL, resulting in an overall level of 68 dB SPL. The duration of each H or I complex stimulus was 0.4 s. In the behavioral experiment, stimulus onset and offset were ramped with a 25 ms Hann window. Stimuli presented to AN fibers were not ramped but still filtered. By analyzing the steady-state response only, any on- and offset effects were excluded.

In both the behavioral experiments and in AN-fiber recordings, three combinations of $f_0$ and $f_c$ were used: $f_c$ = 1600 Hz and $f_0$=200 Hz, $f_c$ = 1600 Hz and $f_0$ = 400 Hz, and $f_c$ = 3200 Hz and $f_0$=400 Hz. The combination of $f_0$=200 Hz and $f_c$ = 3200 Hz was not used because a pilot behavioral test showed that two young untreated gerbils were not able to discriminate I and H even for the largest value of $\Delta f$.

The harmonic rank, *N*, of a condition describes the rank of the center frequency $f_c$ in the harmonic tone complex. The highest value for $\Delta f$ was 42.5% of the fundamental $f_0$. Lower values for $\Delta f$ were computed by dividing 42.5% by 2 n, with n being 1 or increasing integers. The smallest $\Delta f$ presented in the behavioral tests was 0.33% $f_0$. In old gerbils tested behaviorally, additional test stimuli with $\Delta f$=31.87% $f_0$ and 15.94% $f_0$ were presented to ensure that old gerbils were presented with a sufficient number of trials allowing a salient percept for discrimination.

In recordings from gerbil AN fibers, only subsets of the stimuli used in behavior were presented to save time and minimize the number of incomplete datasets due to deterioration or loss of single-unit isolation. The frequency shifts were further grouped into either 'high' (10.6, 15.9, 21.25, 31.8, and 42.5 Hz) or 'low' (0.66, 1.3, and 5.31 Hz), and the stimuli were presented in three different blocks per stimulus condition. The first block comprised the H and two I complex stimuli, one with a high and one with a low shift. The second block comprised I complex stimuli, two high shifts, and another low shift. The third block always comprised the remaining shifts (1.3, 10.6, and 42.5 Hz). The order of stimuli within the blocks was random between fibers. A silent period of 0.6 s occurred between stimuli. This organization of stimuli ensured that the H complex was always presented and could be compared to one low and one high shift. The second and third blocks of stimuli were recorded with declining success.

The periodic envelopes of H and I complexes created in the way described above are similar and offer no cues for discrimination, whereas the TFS of H and I complexes is not identical (*Figure 11A–D*). The TFS of I complexes changes across sequential periods, while the TFS of the H complex does not; only this difference can be used for discrimination. The small random change in the TFS and envelope of both H and I due to the addition of pink noise offers no discrimination cues. The bandpass filtering of the tone complexes served to minimize cochlear excitation pattern cues.

## Experimentally induced synaptopathy using ouabain

For experiments including ouabain treatment, gerbils were anesthetized with ketamine (Ketamin 10%, Bela-pharm GmbH, Vechta, Germany; 135 mg/kg body weight) and xylazine (Xylazin 2%, Ceva Tiergesundheit GmbH, Düsseldorf, Germany; 6 mg/kg body weight) mixed with 0.9% saline injected intraperitoneally. Maintenance doses of one-third of the initial dose were given hourly or as needed. Depth of anesthesia was monitored by withdrawal reflex to toe pinches and via an ECG with needle electrodes in the front- and contralateral hind leg, continuously displayed on an oscilloscope (SDS 1102CNL, SIGLENT Technologies, Hamburg, Germany). Body temperature was continuously controlled with a homeothermic blanket (Harvard Apparatus, Holliston, MA, USA). Experiments were carried out in a sound-attenuating booth. The gerbil's head was fixed within a bite-bar (David Kopf Instruments, Tujunga, CA, USA). Pure, moisturized oxygen (1.5 l/min) was delivered from a tube pointing towards the nose.

Twenty-two gerbils (11 M, 11 F, aged 3–8 months at the beginning of treatment) received either ouabain or artificial perilymph solution. Twenty-four hours before the oubain or artificial perilymph injection, gerbils were treated with antibiotics (oral; Baytril 0.5%, Bayer Animal Health GmbH, Monheim am Rhein, Germany; 0.3 ml per gerbil/day) and a non-steroidal antiphlogistic/analgetic agent (oral; Meloxidyl 0.5 mg/ml, CevaTiergesundheit GmbH; 0.3 ml per gerbil/day; or Novalgin; oral; 500 mg/ml, Sanofi-Aventis Deutschland GmbH, Frankfurt, Germany, 0.21 ml per gerbil/day). The

antibiotic treatment was continued for nine more days, the antiphlogistic/analgetic agent for at least three days. Additionally, gerbils received an antiemetic (oral; Emeprid, Ceva Tiergesundheit GmbH; 0.21 ml per gerbil/day) and, if needed, also sterofundin (4 ml s.c.; B. Braun SE, Melsungen, Germany) or amynin (4 ml s.c.; Boehringer Ingelheim Pharma GmbH Co KG, Ingelheim am Rhein, Germany) as infusion solutions if gerbils were dehydrated, all post-operatively.

Ouabain treatment was carried out according to *Bourien et al., 2014*. After gerbils were anesthetized, they received a subcutaneous infusion with Sterofundin (4 ml). The location where the head-holder met the skin of the face was treated with 2% xylocaine ointment (Lidocaine hydrochloride, Aspen Pharma Trading Limited, Dublin, Ireland). The skin covering the bulla was disinfected with 70% ethanol. A small incision was placed behind the ear to access the bulla. A hole was drilled into the bone with an angled blade and subsequently widened with forceps to reach the round window. A syringe needle tip was placed onto the round window to apply one of the following solutions onto the round window membrane: Eight gerbils were treated with 40 µM ouabain (Ouabain octahydrate, Sigma-Aldrich, Saint Louis, MO, USA), ten with 70 µM ouabain, dissolved in artificial perilymph solution (137 mM NaCl, 5 mM KCl, 2 mM CaCl2, 1 mM MgCl2, 1 mM NaHCO3, and 11 mM glucose; pH 7.4). Four gerbils received only artificial perilymph solution (surgery-only treatment). In 15 gerbils, the solution was continuously delivered by a pump (150 µl for 30 Min). In seven gerbils, the round window niche was bathed with the solution by manually operating the syringe, and after 10 min, the solution was aspirated and reapplied. This procedure was repeated three times. Surgery was concluded by suturing the skin incision over the open middle ear. Gerbils were binaurally treated (n=19), except those that were destined for electrophysiology only, which were monaurally treated (n=3).

Note that the experimenter conducting the electrophysiology and histology was blinded to the type of treatment: the solution that was transferred to the round window was aliquoted by another person. Only after the synapse counts were completed, the treatment was revealed.

## Counting inner-hair-cell synapses
### Immunohistochemistry
Processing of cochleae was carried out as described in detail in *Steenken et al., 2022a*. After concluding the single-unit recordings, gerbils were euthanized with an overdose of pentobarbital (i.p.; Narcoren, Merial GmbH, Hallbergmoos, Germany, 480 mg/kg body weight). Twenty gerbils were cardiovascularly perfused with a mixture of phosphate-buffered saline (PBS; 137 mM NaCl, 10 mM phosphate, and 2.7 mM KCl, pH 7.4) followed by 4% paraformaldehyde in PBS. To prevent blood coagulation, in 15 of these gerbils, heparin (ratiopharm, 25,000 IE/5 ml; 0.2 ml/100 ml) was added to PBS. In 34 gerbils, the bulla was rapidly exposed after euthanasia, and small openings were carefully created at the apex and base of the bony cochlear walls. Cochleae were then fixed in 4% paraformaldehyde in PBS for two days on a shaker at 8 °C. After fixation, all cochleae were decalcified using 0.5 M ethylenediaminetetraacetic acid for two days. Subsequently, cochleae were treated with 1% Triton X-100-PBS for 1 hr to permeabilize the tissue and were then rinsed with 0.2% Triton X-100-PBS. To block unspecific binding sites, cochleae were then incubated in 3% bovine serum albumin (BSA, +0.2% Triton X-100-PBS) blocking solution for 1 hr at room temperature.

Antibodies to label hair cells (anti-MyosinVIIa, IgG polyclonal rabbit; Proteus Biosciences, Ramona, CA, USA; diluted 1:400), presynaptic ribbons (anti-CtBP2 (C-terminal binding protein) antibody, IgG1 monoclonal mouse; BD Biosciences, Eysins, Switzerland; diluted 1:400), and postsynaptic receptor patches (anti-GluR2a antibody, IgG2a monoclonal mouse; Millipore, Burlington, MA, USA; diluted 1:200) in blocking solution were used. Cochleae were incubated in this mixture at 37 °C overnight. Next, cochleae were again rinsed in 0.2% Triton X-100-PBS.

Secondary antibodies were chosen to match the hosts of the primary antibodies: goat anti-mouse (IgG1)-AF488 (Molecular Probes Inc, Eugene, OR, USA; diluted 1:1000), goat anti-mouse (IgG2a)-AF568 (Invitrogen, Carlsbad, CA, USA; diluted 1:500), and donkey anti-rabbit (IgG)-AF647 (Molecular Probes Inc; diluted 1:1000). Secondary antibodies were diluted in blocking solution and cochleae were incubated in this solution at 37 °C overnight and subsequently rinsed in PBS.

After the immunostaining, a subset of the cochleae from untreated animals older than 12 months received treatment with an auto-fluorescence quencher (n=13; TrueBlack Lipofuscin Autofluorescence Quencher, 20 x in DMF, Biotum, Hayward, CA, USA). These cochleae were cut in half and incubated in a 5% true black-70% ethanol mixture for 1 min and then rinsed in PBS. Finally, all cochleae

were micro-dissected under a stereomicroscope and the resulting 5–11 organ-of-Corti pieces were mounted on slides using Vectashield Mounting Medium (Vector Laboratories, Burlingame, CA, USA, H-1000).

## Image acquisition

Overview images of cochlear pieces were acquired with a light microscope (Nikon 90i with NIS Elements software, Version 4.30, Nikon, Minato, Tokyo, Japan) using a 4 x objective. After measuring the length of the cochlea as a line along the row of IHCs with the NIS software, the cochlear location at 3.8 mm from the apex, corresponding to 2 kHz (*Müller, 1996*), was chosen, because 2 kHz was part of all stimulus conditions (data for additional locations corresponding to 4, 8, and 16 kHz are also provided in *Figure 1*). Images were taken at this position with a confocal microscope (Leica Microsystem CMS GmbH, Wetzlar, Germany, Leica TCS SP8 system) using an oil-immersion objective (40 x, numerical aperture 1.3). Lasers (488 nm and 522 nm optically pumped semiconductor laser), or a 638 nm diode, respectively, excited the different fluorescence tags, and released photons were counted with a hybrid detector. Image stacks had voxel dimensions of 0.05–0.0998 µm (XY) and 0.3 µm (Z). All confocal stacks were deconvolved (Huygens Essentials, Version 15.10, SVI, Hilversum, Netherlands) with default settings (maximum iteration: 80; signal-to-noise ratio: 10; quality threshold: 0.01), using a theoretical point-spread function.

Synapses were counted using ImageJ (FIJI, *Schindelin et al., 2012*). For each cochlear position, the number of functional synapses was evaluated for 5 IHCs by manually counting the co-localized ribbons and glutamate patches. Contrast and brightness were manually adjusted for each stack.

## Evoked potential recordings

Hearing sensitivity of all gerbils was evaluated prior to and during AN recordings using ABRs and CAPs. For ABRs, a platinum needle electrode was placed subcutaneously near the ipsilateral mastoid. For CAPs, a custom-made silver-wire ball electrode was placed at the round window niche after opening the bulla. For both measurements, the reference electrode was placed in the neck, and a ground electrode was placed on one hind leg. The stimulus presentation was controlled via a custom-written MATLAB software (https://github.com/rainbeu/abr-software, copy archived at *Beutelmann, 2026*) and delivered through an RME Hammerfall DSP Multiface II sound card (RME Audio, Haimhausen, Germany). The stimuli were amplified (HB7, TDT Inc) and presented to the gerbil's ear via earphones (IE 800, Sennheiser, Wedemark, Germany) attached to an ear-bar.

Evoked potentials were amplified ($\times 10^3$) and band-pass filtered (ABR: 0.3–3 kHz; CAP: 0.005–10 kHz) using an ISO-80 pre-amplifier (World Precision Instruments, Sarasota, FL, USA), and sampled using the digital signal processor (Hammerfall DSP Multiface II, RME Audio; 48 kHz sampling rate) controlled by custom MATLAB software (*Beutelmann, 2026*).

Prior to ouabain treatment (CAP), and again prior to AN recordings (ABR and CAP), responses to chirps were recorded at a range of levels, from below threshold up to 20 dB above threshold, in steps of 5 dB, and averaged over 300 repetitions. During single-unit recording sessions, ABR thresholds were re-checked periodically and visually evaluated, and the experiment was terminated if threshold deteriorated by more than 30 dB. For post-hoc analysis, CAP amplitudes (N1 – P1) were determined with the use of a custom-written Matlab script (*Beutelmann, 2026*). Threshold was defined as the first level step that exceeded a 2 µV CAP.

## Single-unit recordings

### Stimulus presentation

Stimuli for single-unit recordings were generated with custom-written MATLAB scripts (*Beutelmann, 2026*) controlling a digital signal processor (RX6, TDT Inc, Alachua, FL, USA) and an attenuator (PA5, TDT Inc). The signal was then routed through a headphone buffer (HB7, TDT Inc) and presented via earphones (IE 800, Sennheiser, Wedemark, Germany) attached to an ear-bar. Stimuli were calibrated at the start of each experiment using a probe-tube microphone (ER7-C, Etymotic Research Inc, Elk Grove Village, IL, USA) attached to the ear-bar and a microphone amplifier (MA3, TDT Inc, 40 dB amplification).

## Single-unit recordings

Single-unit data was obtained in 27 gerbils (see *Table 2*). After the gerbil was anesthetized, a single dose of non-steroidal antiphlogistic agent (meloxicam, 0.2 mg/kg, Boehringer Ingelheim Pharma GmbH Co KG) was administered subcutaneously. The head of the gerbil was oriented with a bite bar and firmly fixed to the setup via a screw glued with dental cement to the exposed skull. The pinna of one ear was removed to place an ear bar directly at the bony edge of the ear canal. The ear bar was sealed with petroleum jelly to the bone to form a closed sound system. To prevent pressure buildup in the middle ear, a small hole was drilled into the dorsal bulla with an angled blade.

The AN was accessed via a dorsal approach. The skin covering the occipital bone and the bone itself were removed and the lateral part of the cerebellum was aspirated. The brainstem was left intact and was gently pushed medially using tiny saline-soaked cotton balls to access the proximal part of the eighth cranial nerve. A glass electrode (GB120F-10, Science Products GmbH, Hofheim am Taunus, Germany; pulled using a P-2000, Sutter Instruments Co., Novato, CA, USA), filled with 3 M KCl solution and with a resistance of ~10–30 MΩ, was placed above the AN under visual control. The electrode was advanced through the AN via a remote-controlled piezo motor (Burleigh 6000 ULN inchworm motor controller and 6005 ULN handset, Burleigh, Inc, Fishers, NY, USA). Recordings were amplified (WPI 767, World Precision Instruments Inc, Sarasota, FL, USA), filtered for line-frequency noise (50 Hz, Hum Bug, Quest Scientific Instruments Inc, North Vancouver, BC, Canada), and digitized (sampling rate: 48,828 Hz; RX6, TDT Inc, Alachua, FL, USA). Additionally, the analog signal was guided to an audio monitor (MS2, TDT Inc) and displayed on an oscilloscope (SDS 1102CNL, SIGLENT Technologies, Hamburg, Germany).

Search stimuli (broadband noise, 1–9 kHz at 50 dB SPL) were played while the electrode was advanced into the AN. When a unit was isolated, its best frequency (BF) was first assessed audio-visually using 50 ms duration tones. Next, a response curve was obtained by presenting pure tone bursts (50 ms duration, including 5 ms raised-cosine rise/fall times, 0–30 dB above threshold, three repetitions) at a range of frequencies within approximately ± 1 octave around the audio-visually determined BF. Subsequently, tone bursts at BF were presented over a range of levels (10–79 dB SPL, 50 ms duration including 5 ms raised-cosine rise/fall times, 10 repetitions) to derive a rate-level function. At randomly inserted trials, no stimulus was presented to determine the spontaneous rate (SR). Subsequently, TFS1 stimuli were presented at a fixed level of 60 dB SPL.

Spike detection from the recorded signal was conducted offline with a custom-written MATLAB script (*Steenken et al., 2022b*). In short, the bandpass-filtered (300–3000 Hz) recordings were manually screened for threshold-crossing events. The threshold could be adjusted on a trial-by-trial basis to compensate for variations in spike amplitude. The time of each spike's peak amplitude was saved as the spike time. Spike trains were excluded if: (1) more than 10% of the spikes for a given stimulus condition were judged to be below the set threshold criterion, or (2) any inter-spike intervals <0.6 ms occurred, indicating inadequate unit isolation (*Heil et al., 2007*), or (3) a pre-potential was present in the spike waveform, suggesting a recording from the ventral cochlear nucleus (*Keine and Rübsamen, 2015*), or (4) the rate-level function was non-monotonic, also suggesting a potential cochlear nucleus recording (e.g. *Davis et al., 1996*; *Joris et al., 1994*; *Sinex et al., 2001*).

## Neural recordings - Data analysis

BF was defined as the frequency that evoked the highest mean discharge rate in the recorded response curve. The threshold at BF was determined from the rate-level curve as the lowest stimulus level where the firing rate was higher than mean spontaneous rate SR +1.2 times the standard deviation of spontaneous rate SR and higher than 15 spikes/s. The bandwidth of the response curve, at 50% of the maximal driven rate, was taken as a measure of neural tuning, but only if the response curve had been obtained at ≤20 dB above threshold.

A number of measures were derived from the AN responses to TFS1 stimuli: (a) Peri-stimulus time histograms (PSTHs), (b) rate histograms, and (c) vector strengths (VS) corresponding to certain periodicities in the stimuli. Only the 375 ms long steady-state part of the responses, excluding a 12.5 xms onset and offset fringe, respectively, was analyzed.

First, peri-stimulus time histograms (PSTHs) were calculated for each animal/fiber/condition/Δf combination, pooling over repeated stimulus presentations within each combination. Spike times were collected into histograms using bin widths of 1/10 of the $f_0$ period (0.5 ms for 200 Hz and 0.25

ms for 400 Hz). The histogram counts were converted into instantaneous spike rates by dividing by the bin width and by the number of stimulus repetitions.

Second, average rate and SR histograms were calculated for each group/condition/Δf combination, pooling over repeated stimulus presentations, animals, and fibers within each combination. Average rates were computed over the 375 ms steady-state portion of the response. SR was calculated from trials without stimulation. The driven rate was calculated as the difference between stimulated absolute rate and SR. Average driven rates and SRs were collected into histograms using bin widths of 2.5 spikes per second and five spikes per second, respectively. The average-driven-rate histogram counts were divided by the number of spike trains to convert the counts to relative frequency and smoothed with a moving-average window of three bin lengths.

Third, VS frequency spectra were calculated for each group/condition/Δf combination, pooling over repeated stimulus presentations, animals, and fibers within each combination. For this, all-order inter-spike intervals (ISIs) were calculated from each spike train and collected into histograms using a bin width of 50 µs, corresponding to a sampling rate of 20 kHz. The bin positions ranged from –375 ms to +375 ms. A zero-centered Gaussian window with a standard deviation of 200 ms was multiplied with the histogram to suppress fringe effects and spurious side lobes in the spectra. The histogram counts were divided by the total number of ISIs. A subsequent fast Fourier transform of the histograms resulted in the ISI power spectrum, and the square root of the spectral magnitudes was identical to the VS at each bin center frequency. The spectral resolution was 1.33 Hz.

VS single-peak data was calculated for each animal/fiber/condition/Δf combination, pooling over repeated stimulus presentations within each combination. In contrast to the spectra, the peak VS values were calculated individually for each relevant frequency $f$:

$$V(f) = \left| \frac{1}{N} \sum_{i=1}^{N} e^{-i2\pi f t_i} \right|$$

where $t_i$ are the spike times, and $N$ is the number of spikes in each spike train. For significance analysis of the VS, the Rayleigh statistic $z$ (henceforth referred to as $z$-value) was calculated: $z = N_{avg} \cdot V_{avg}^2(f)$, where $N_{avg}$ is the average number of spikes per spike train and $V_{avg}$ is the average VS across stimulus presentations. The Rayleigh $z$ statistic has useful properties for statistical analyses because VS values resulting from higher numbers of spikes have larger weights, and logarithms of ratios of $z$-values are approximately normally distributed, as opposed to VS ratios or differences.

As a metric for TFS representation, the log ratio between two $z$-values was derived: First, from responses to the H complex, the $z$-value, $z_0$, at the frequency component that the fibers on average responded to maximally, $f_{maxpeak}$, was obtained. Second, this value was divided by the $z$-value, $z_{\Delta f}$, at the shifted frequency of the I complex, also in the maximal average response range (filled circles in *Figure 12*). This ratio was used as the dependent variable in the first ANOVA model and reflects high temporal resolution; large positive values correspond to stronger phase-locked responses to the TFS of the stimulus. This metric was termed the TFS log-z-ratio.

Another log ratio between $z$-values was used as a metric for the balance between envelope and TFS representation (crosses in *Figure 12*): The (harmonic, unshifted) $z_0$ value at the fundamental frequency, $f_0$, was divided by the (inharmonic, shifted) $z_{\Delta f}$ value in the maximal average response range, $f_{maxpeak}$. Large positive log ratios indicate a stronger response to the stimulus envelope, whereas large negative log ratios indicate a stronger response to the stimulus TFS. This metric was termed ENV/TFS log-z-ratio.

The representation of Δf shift in the responses was also tested by calculating $z$-values at the different harmonic frequencies, combined with all possible Δf shifts. The actual position of the maximum $z$-value in terms of Δf and the corresponding $z$-value is summarized in *Figure 5* for two TFS peak frequencies (to which the Δf shifts were added): at $f_0$ of the corresponding condition and at 1 kHz for the 200/1600 Hz condition, at 800 Hz for the 400/1600 Hz condition, and at 2 kHz for the 400/3200 Hz condition, respectively. The latter TFS peak frequencies were identified as the harmonics actually present in the acoustic stimulus, which had the maximum $z$-value in the summarized spectra (*Figure 5*).

For the single-unit data, several main factors were defined. The low-SR fiber class comprised fibers with SR ≤18 spikes/s, and the high-SR class comprised fibers with SR >18 spikes/s. In addition, fibers in the low-BF class had BF ≤1850 Hz, and fibers in the high-BF class had BF >1850 Hz.

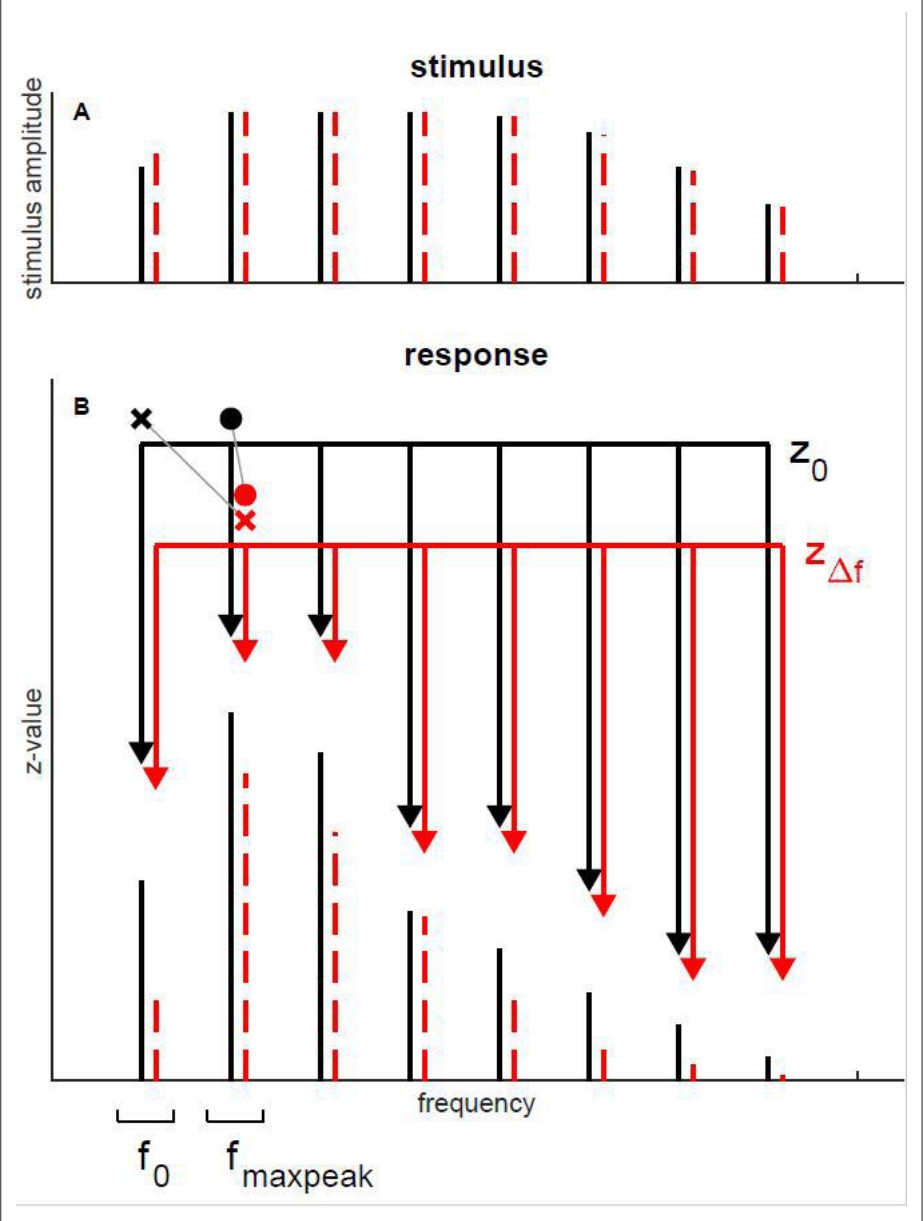

**Figure 12.** Schematic representation of statistical contrasts. Schematic of the frequency spectrum of a harmonic (H, black solid lines) and inharmonic (I, red dashed lines) TFS1 stimulus in panel (**A**) and schematic representation of the phase-locking spectrum in a neuronal response in panel (**B**). Black arrows and symbols point to the harmonic, unshifted frequencies, termed 'z$_0$' and red arrows and symbols point to the inharmonic, shifted frequencies, termed 'z$_{df}$.' The annotated brackets below the frequency axis mark representative regions of the fundamental frequency, f$_0$, and of the maximal average response, f$_{maxpeak}$. Filled connected circles show the log-z-ratio used for assessing the temporal fine structure (TFS) representation, connected crosses show the log-z-ratio used for assessing the balance between f$_0$/ENV and TFS representation.

## Behavioral procedure

The behavioral experiments were conducted in a single-walled sound-attenuating chamber (Industrial Acoustics, Type 401 A, Industrial Acoustic Company GmbH, Mönchengladbach, Germany) lined with a 15 cm thick layer of sound- absorbing foam (Pinta Acoustics Pyramide 100/50 on Pinta Acustics PLANO Type 50/0, Seyboth & Co., Regensburg, Germany). The walls and all devices within the chamber produced no relevant sound reflections. The time for the reverberation in the chamber to

decay by 60 dB was 12 ms, indicating nearly anechoic conditions. These conditions allowed faithful presentation of the TFS1 stimuli with the desired acoustic cues in the free sound field.

In the center of the chamber, a 30-cm-long wire-mesh platform was located at 90 cm height, with an elevated pedestal at its center where the gerbils waited during the measurement. This construction minimized sound reflections. At one end of the platform, a food bowl was located, attached via a tube to a custom-made feeder that did not obstruct the sound path. Gerbils were rewarded for correct discrimination with a 10 mg custom-made food pellet (Altromin International, type 1324 rodent pellets enriched with sunflower oil and spelt flour; Altromin Spezialfutter GmbH & Co. KG, Lage, Germany). A loudspeaker (Canton Plus XS, frequency range: 150 Hz - 21 kHz, Canton, Weilrod, Germany) was mounted 30 cm in front of the pedestal to the side of the food bowl, at 0° elevation and azimuth relative to the gerbil's normal head position when sitting on the pedestal. Out of the gerbil's reach, a system of two custom-made light barriers was installed to detect the gerbil's position and facing direction on the pedestal. An infrared camera (Conrad 150001 C-MOS camera module, Conrad Electronics, Hirschau, Germany) was positioned above the platform to observe the gerbil's movements on the platform under invisible infrared light. The stimulus presentation, registration of light barriers switching, and feeder were controlled by custom-written software on a Linux-based PC with an RME sound card (Hammerfall DSP Multiface II, sampling frequency 48 kHz). The signal output from the sound card was manually attenuated (Texio type RA-902A, TEXIO Technology Corporation, Kanagawa, Japan), amplified (Rotel type RMB 1506, Rotel Tokyo, Japan), and presented by the loudspeaker. Before testing started on each day, the system was calibrated (±1 dB) with a sound level meter (Brüel and Kjaer type 2238 Mediator, Hottinger Brüel & Kjær, Virum, Denmark) positioned on the elevated pedestal at about the position of the gerbil's head.

The behavioral experiment used an operant Go/No-Go paradigm. Gerbils were trained to wait on the pedestal facing in the direction of the loudspeaker. During the whole session, the H complex was played every 1.3 s as a reference stimulus. As soon as the gerbil jumped onto the pedestal facing the loudspeaker, a trial was started with a random waiting time between 2 and 7 s. When the waiting time had elapsed, a target stimulus, the I complex, was played instead of the reference H complex. The target stimuli could either be equal to the reference H complex (sham trial) for estimating the false-alarm rate or an I complex (test trial) for testing discrimination ability. Leaving the pedestal before the waiting time elapsed resulted in the restart of the trial with a new waiting time. Leaving the pedestal within 1.3 s after the onset of the target stimulus of a test trial, as indicated by the light barriers, was registered as a 'hit' and resulted in a food reward. Staying on the pedestal in a test trial was counted as a 'miss' with no food reward. The animal had to leave the pedestal and jump back onto it in the correct sequence to start a new trial after a 'miss' occurred. Leaving the pedestal within 1.3 s after the start of the target stimulus in a sham trial was registered as a 'false alarm,' while staying on the pedestal during a sham trial was registered as a 'correct rejection.' No food reward was provided in sham trials. To keep the gerbils motivated during the session, a salient test trial not included in the analysis was inserted after each correct rejection.

Each session contained 10 blocks with nine trials each (six test trials, three sham trials). All stimuli within a given block were from the same condition (200/1600 Hz, 400/1600 Hz, or 400/3200 Hz). The first block of a session was a warm-up block with Δf=42.5% that was not included in the analysis. To interleave simple and more difficult conditions, blocks of the different conditions were pseudo-randomly distributed throughout the session. The least salient condition (200/1600 Hz) did not occur within two consecutive blocks. Additionally, the sham and test trials within each block were pseudo-randomly distributed with the constraint that no sham trials followed one another within and across blocks. Data collection started when the gerbil achieved a false alarm rate below 20% in three consecutive sessions with hit rates for the three highest Δf values that would allow determining a discrimination threshold.

Sessions were included in the analysis if the gerbil completed all 90 trials and the false alarm rate did not exceed 20%. Data collection continued until the sample size for each condition and Δf value was at least 20 trials. The sensitivity for each Δf value was defined by applying the z-transform, $d'=z(\text{hit rate}) - z(\text{false-alarm rate})$. Individual discrimination thresholds for each animal and condition were then calculated from sensitivity, $d'$, as a function of inharmonic frequency shift, Δf%. A threshold at $d'=1$ was obtained by linearly interpolating between the first Δf% shift where $d'>1$ and the previous Δf% shift with $d' \leq 1$. If all $d'$ values were below 1, a linear fit was estimated through all $d'$ values as

a function of Δf% shifts and the threshold was extrapolated at $d'$=1. If all $d'$ values were larger than 1, the threshold was set to half of the minimal Δf% shift measured in the specific experiment. If none of the above was possible and either all $d'$<1, the fitted slope was not positive, or the extrapolated threshold was larger than 100% shift (equivalent to $Δf=f_0$ at threshold), the final threshold was limited to 100% (5 out of 72 thresholds).

## Statistical analysis

All analyses of variance were calculated with IBM SPSS Statistics for Windows, Version 29.0 (IBM Corp, Armonk, NY). The family-wise error rate of all multiple post-hoc comparisons was controlled by Bonferroni correction of the significance levels.

## Acknowledgements

This work was supported by the DFG priority program 'PP 1608,' the DFG Cluster of Excellence EXC 1077/1 'Hearing4all' (Project ID 390895286). We thank Laurel Carney for helpful comments on a previous version of this manuscript. Susanne Groß for assistance with the behavioral experiments. Amarins N Heeringa for providing analysis scripts for AN data, analyzing, and curation of electrophysiological data. Nadine Thiele for generous assistance with the gerbil care and for implementing the blinding process. Furthermore, many thanks go to Sonja Standfest for the great assistance with gerbil care. Sonny Bovee is acknowledged for processing cochleae, Julia Forst, and Carina Lützow for assistance with the spike detection and immunostainings, and Julia Winter for assistance with the immunostainings. We acknowledge the Fluorescence Microscopy Service Unit, Carl von Ossietzky University of Oldenburg, for the use of the imaging facilities.

## Additional information

### Funding

| Funder | Grant reference number | Author |
|---|---|---|
| Deutsche Forschungsgemeinschaft | PP 1608 | Christine Köppl Georg M Klump |
| Deutsche Forschungsgemeinschaft | EXC 1077/1 "Hearing4all" (Project ID 390895286) | Christine Köppl Georg M Klump |

The funders had no role in study design, data collection and interpretation, or the decision to submit the work for publication.

### Author contributions

Friederike Steenken, Formal analysis, Investigation, Visualization, Writing – original draft, Writing – review and editing; Rainer Beutelmann, Software, Formal analysis, Visualization, Writing – original draft, Writing – review and editing; Henning Oetjen, Formal analysis, Investigation, Writing – original draft; Christine Köppl, Conceptualization, Supervision, Writing – original draft, Writing – review and editing; Georg M Klump, Conceptualization, Formal analysis, Supervision, Funding acquisition, Writing – original draft, Project administration, Writing – review and editing

### Author ORCIDs

Rainer Beutelmann ⬤ https://orcid.org/0000-0001-9708-4113
Christine Köppl ⬤ https://orcid.org/0000-0003-0729-709X
Georg M Klump ⬤ https://orcid.org/0000-0003-2501-8090

### Ethics

All protocols and procedures were approved by the Niedersächsisches Landesamt für Verbraucherschutz und Lebensmittelsicherheit (LAVES), Germany, permit AZ 33.19-42502-04-15/1990. All procedures were performed in compliance with the NIH Guide on Methods and Welfare Consideration in Behavioral Research with Animals (National Institute of Mental Health, 2002).

Reviewer #1 (Public review): https://doi.org/10.7554/eLife.102890.3.sa1
Reviewer #2 (Public review): https://doi.org/10.7554/eLife.102890.3.sa2
Reviewer #3 (Public review): https://doi.org/10.7554/eLife.102890.3.sa3
Author response https://doi.org/10.7554/eLife.102890.3.sa4

## Additional files

### Supplementary files
MDAR checklist

### Data availability
The data that are presented in the figures and were statistically analyzed are uploaded to a Zenodo repository (https://doi.org/10.5281/zenodo.15546625). We have deposited the software package developed by the coauthor Rainer Beutelmann in GitHub (https://github.com/rainbeu/abr-software, copy archived at *Beutelmann, 2026*).

The following dataset was generated:

| Author(s) | Year | Dataset title | Dataset URL | Database and Identifier |
|---|---|---|---|---|
| Klump GM, Beutelmann R | 2025 | Original raw data files for Elife publication Auditory perception and neural representation of temporal features are altered by age but not by cochlear synaptopathy | https://doi.org/10.5281/zenodo.15546625 | Zenodo, 10.5281/zenodo.15546625 |

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

## Appendix 1

This appendix summarizes extended statistical results that complement the results reported in the main manuscript.

Below, we list the full ANOVA results, including the principal effects that are reported in the main manuscript. The factors used in the ANOVAs reported here had the following factor levels:

- **Group:** young-adult, sham, ouabain-high, old
- **Condition:** 200/1600, 400/1600, 400/3200 ($f_0/f_c$ in Hz)
- **BF class:** low: BF ≤1850 Hz, high: BF >1850 Hz
- **SR class:** low: SR ≤18 spikes/s, high: SR >18 spikes/s
- **Δf%:** as a covariate ranging from 0 to 21.25%

All significances are reported at the 5% level, with Bonferroni correction for post-hoc tests and planned comparisons.

### Effects on mean driven rate supplemental to 'Average rate does not carry sufficient information about inharmonic frequency shift'

Note that driven rate (mean rate - spontaneous rate) was used here. The differences found below are consistent with known differences between low-SR (higher driven rates) and high-SR fibers (lower driven rates; *Steenken et al., 2021*), and with expected differences across BF classes (low-BF fibers less driven by stimuli with $f_c$ = 3200 Hz and high-BF fibers less driven by stimuli with $f_c$ = 1600 Hz). Differences between groups are also consistent with the sample distributions shown in *Figures 1 and 4A*

- **Significant main effects:**
  - **Group:** $F(3,943)=10.017$, $p=1.676 \times 10^{-6}$
    - Mean driven rate was lower in the sham group than in any other group.
  - **BF class:** $F(1,943)=6.297$, $p=0.0123$
    - Mean driven rate was lower for high BF class (BF >1850 Hz).
- **Significant two-way interactions:**
  - **Condition × group:** $F(6,943)=2.525$, $p=0.0198$
    - Planned comparisons between groups within each condition, and between conditions within each group: In the 400/3200 condition, the mean driven rate of the sham group was significantly lower than in any other group. The mean driven rate in the sham group was significantly lower than in the young-adult group for each condition.
  - **Condition × BF class:** $F(2,943)=10.562$, $p=2.909 \times 10^{-5}$
    - Planned comparisons between conditions within each BF class, and between BF classes within each condition: In the x/1600 conditions, the mean driven rate in the high-BF class was significantly lower than in the low-BF class. No significant differences between conditions within each BF class.
  - **Group × BF class:** $F(3,943)=10.189$, $p=1.316 \times 10^{-6}$
    - Planned comparisons between groups within each BF class, and between BF classes within each group: In the low-BF class, the mean driven rate in the sham group was significantly lower than in all other groups. For the ouabain-high and old groups, the mean driven rate in the high-BF class was lower than in the low-BF class.
  - **Group × SR class:** $F(2,943)=8.148$, $p=3.103 \times 10^{-4}$
    - Planned comparisons between groups within each SR class, and between SR classes within each group: In the young-adult group, the mean driven rate in the high-SR class was lower than in the low-SR class. In the high-SR class, the mean driven rate in the sham group was lower than in any other group. (No samples for the sham group in the low-SR class).
- **Non-significant main effects and interactions:**
  - **Δf%:** *F(1,943)=0.018, p=0.8933 (reported in main manuscript)*
  - **Condition:** $F(2,943)=2.187$, $p=0.1129$
  - **SR class:** $F(1,943)=1.884$, $p=0.1702$
  - **Group × Δf%:** $F(3,943)=0.394$, $p=0.7575$
  - **BF class × Δf%:** $F(1,943)=0.052$, $p=0.8195$
  - **Condition × Δf%:** $F(2,943)=0.081$, $p=0.9219$

- **SR class × Δf%:** F(1,943)=0.002, *p*=0.9657
- **Condition × SR class:** F(2,943)=0.698, *p*=0.4977
- **BF class × SR class:** F(1,943)=0.388, *p*=0.5336

## Effects on TFS representation (TFS log-z-ratio) supplemental to 'Neural representation of TFS was not degraded in AN fibers of old or synaptopathic gerbils'

A high TFS log-z-ratio indicates strong phase locking to the stimulus TFS.

- **Significant main effects:**
  - *Δf%: F(1,943)=104.403, p=2.569 × 10⁻²³ (reported in main manuscript)*
  - *Condition: F(2,943)=7.495, p=5.894 × 10⁻⁴ (reported in main manuscript)*
    - TFS log-z-ratio was significantly lower in the 400/3200 condition, compared to both the 200/1600 and 400/1600 conditions.
- **Significant two-way interactions:**
  - *Condition × Δf%: F(2,943)=20.292, p=2.353 × 10⁻⁹ (reported in main manuscript)*
    - Based on the parameter estimates of the ANOVA model: The slope of the TFS log-z-ratio vs. Δf% was significantly different from 0, for the 200/1600 and 400/1600 conditions. It was significantly higher in the 400/1600 condition, compared to the 200/1600 condition.
  - *BF class × Δf%: F(1,943)=31.05, p=3.282 × 10⁻⁸ (reported in main manuscript)*
    - Based on the parameter estimates of the ANOVA model: The slope of the TFS log-z-ratio vs. Δf% was significantly higher in the low-BF class than in the high-BF class.
  - **Condition × group:** F(6,943)=7.583, *p*=5.808 × 10⁻⁸
    - Planned comparisons between groups within each condition, and between conditions within each group: In the 200/1600 condition, TFS log-z-ratio in the ouabain-high group was significantly higher than in the old group. For all groups except the old group, TFS log-z-ratio in the 400/1600 condition was significantly higher than in the 400/3200 condition. In the sham group, the TFS log-z-ratio in the 400/1600 condition was significantly higher than in the 200/1600 condition. In the ouabain-high group, the TFS log-z-ratio in the 200/1600 condition was significantly higher than in the 400/3200 condition.
  - **Condition × BF class:** F(2,943)=5.774, *p*=3.200 × 10⁻³
    - Planned comparisons between BF classes within each condition, and between conditions within each BF class: In the high-BF class, TFS log-z-ratio in the 400/3200 condition was significantly lower than in any other condition. In the 400/3200 condition, the TFS log-z-ratio in the high-BF class was significantly lower than in the low-BF class.
  - **Group × BF class:** F(3,943)=4.228, *p*=5.600 × 10⁻³
    - Planned comparisons between groups within each BF class, and between BF classes within each group: In the high-BF class, the TFS log-z-ratio in the ouabain-high group was significantly higher than in the old group.
  - **BF class × SR class:** F(1,943)=4.918, *p*=0.0268
    - In the high-SR class, TFS log-z-ratio in the low-BF class was significantly higher than in the high-BF class.
- **Non-significant main effects and interactions:**
  - *Group: F(3,943)=1.757, p=0.1537 (reported in main manuscript)*
  - *Group × Δf%: F(3,943)=1.193, p=0.3112 (reported in main manuscript)*
  - **BF class:** F(1,943)=0.103, *p*=0.7479
  - **SR class:** F(1,943)=0.985, *p*=0.3212
  - **Condition × SR class:** F(2,943)=2.302, *p*=0.1006
  - **Group × SR class:** F(2,943)=0.556, *p*=0.5736
  - **SR class × Δf%:** F(1,943)=1.203, *p*=0.2730

## Effects on ENV/TFS representation balance (ENV/TFS log-z-ratio) supplemental to 'Representation of $f_0$ was enhanced in AN fibers of old gerbils'

The ENV/TFS log-z-ratio reflects the relative emphasis in the AN fibers' responses on either TFS (negative ENV/TFS log-z-ratio) or envelope locking (positive ENV/TFS log-z-ratio).

- **Significant main effects:**
  - ***Condition:*** *F(2,943)=43.929, p=5.744 × 10^{-19} (reported in main manuscript)*
    - The 200/1600 and 400/1600 conditions had negative ENV/TFS log-z-ratios, the 400/3200 condition had a positive ENV/TFS log-z-ratio. All were significantly different from each other, with the 200/1600 condition showing the most negative value.
  - ***SR class:*** *F(1,943)=29.138, p=8.533 × 10^{-08} (reported in main manuscript)*
    - In the high-SR class, the ENV/TFS log-z-ratio was negative and significantly different from zero. In the low-SR class, it was not significantly different from zero.
  - ***Δf%:*** *F(1,943)=4.278, p=0.0389 (reported in main manuscript)*
- **Significant two-way interactions:**
  - **Condition × group:** F(6,943)=11.439, *p=2.242 × 10^{-12}* (reported in main manuscript)
    - – Planned comparisons between groups within each condition, and between conditions within each group: In addition to the main condition effect: In the 200/1600 condition, the ENV/TFS log-z-ratio was more negative in the sham group than both in the young-adult group and in the old group. In the 400/3200 condition, the ENV/TFS log-z-ratio was more positive in the old group than in the young-adult group. In the sham group, the ENV/TFS log-z-ratio was more negative in the 200/1600 condition than in the 400/1600 condition.
  - **Group × BF class:** F(3,943)=11.354, *p=2.551 × 10^{-7}*
    - Planned comparisons between groups within each BF class, and between BF classes within each group: In the high-BF class, the ENV/TFS log-z-ratio was significantly more negative in the sham and ouabain-high groups than in the young-adult and old groups.
  - **BF class × SR class:** F(1,943)=5.234, *p=0.0224*
    - In the order from most negative to most positive ENV/TFS log-z-ratio: low-BF/high-SR, high-BF/high-SR, high-BF/low-SR, low-BF/low-SR. Both high-SR values were significantly below zero and different from each other; both low-SR values were not significantly different from zero, and not different from each other.
- **Non-significant main effects and interactions:**
  - **Group:** F(3,943)=0.838, *p=0.4731*
  - **BF class:** F(1,943)=0.439, *p=0.5076*
  - **Group × Δf%:** F(3,943)=0.236, *p=0.8711*
  - **Condition × Δf%:** F(2,943)=1.673, *p=0.1883*
  - **BF class × Δf%:** F(1,943)=0.005, *p=0.9410*
  - **SR class × Δf%:** F(1,943)=0.58, *p=0.4464*
  - **Condition × BF class:** F(2,943)=0.414, *p=0.6611*
  - **Condition × SR class:** F(2,943)=0.522, *p=0.5936*
  - **Group × SR class:** F(2,943)=2.442, *p=0.0875*

