## [Editor Report · eLife Assessment]

This study tested the specific hypothesis that age-related changes to hearing involve a partial loss of synapse connections between sensory cells in the ear and the nerve fibers that carry information about sounds to the brain, and that this interferes with the ability to discriminate rapid temporal fluctuations in sounds. Physiological, behavioral, and histological analyses provide a powerful combination to test this hypothesis in gerbils. Contrary to previous suggestions, it was found that chemically-induced isolated synaptopathy (at similar levels as observed in aged gerbils) did not result in worse performance on a behavioral task measuring sensitivity to temporal fine-structure, nor did it produce degradations in auditory-nerve fiber encoding of fine structure. Aged gerbils showed degraded behavior and stronger than normal envelope responses, but temporal fine-structure coding was not affected; interpreted by the authors as suggesting central processing contributions to aging effects on discrimination. These findings are **important** for advancing our knowledge of the mechanistic bases for age-related changes to hearing, and the evidence provided is **solid** with the results largely supporting the claims made and minor limitations related to possible confounds discussed in reasonable depth.

---

## [Referee Report · Reviewer #1 (Public review)]

Summary:

The authors investigate the effects of aging on auditory system performance in understanding temporal fine structure (TFS), using both behavioral assessments and physiological recordings from the auditory periphery, specifically at the level of the auditory nerve. This dual approach aims to enhance understanding of the mechanisms underlying observed behavioral outcomes. The results indicate that aged animals exhibit deficits in behavioral tasks for distinguishing between harmonic and inharmonic sounds, which is a standard test for TFS coding. However, neural responses at the auditory nerve level do not show significant differences when compared to those in young, normal-hearing animals. The authors suggest that these behavioral deficits in aged animals are likely attributable to dysfunctions in the central auditory system, potentially as a consequence of aging.To further investigate this hypothesis, the study includes an animal group with selective synaptic loss between inner hair cells and auditory nerve fibers, a condition known as cochlear synaptopathy (CS). CS is a pathology associated with aging and is thought to be an early indicator of hearing impairment. Interestingly, animals with selective CS showed physiological and behavioral TFS coding similar to that of the young normal-hearing group, contrasting with the aged group's deficits. Despite histological evidence of significant synaptic loss in the CS group, the study concludes that CS does not appear to affect TFS coding, either behaviorally or physiologically.

Strengths:

This study addresses a critical health concern, enhancing our understanding of mechanisms underlying age-related difficulties in speech intelligibility, even when audiometric thresholds are within normal limits. A major strength of this work is the comprehensive approach, integrating behavioral assessments, auditory nerve (AN) physiology, and histology within the same animal subjects. This approach enhances understanding of the mechanisms underlying the behavioral outcomes and provides confidence in the actual occurrence of synapse loss and its effects.The study carefully manages controlled conditions by including five distinct groups: young normal-hearing animals, aged animals, animals with CS induced through low and high doses, and a sham surgery group. This careful setup strengthens the study's reliability and allows for meaningful comparisons across conditions. Overall, the manuscript is well-structured, with clear and accessible writing that facilitates comprehension of complex concepts.

Weakness:

The stimulus and task employed in this study are very helpful for behavioral research, and using the same stimulus setup for physiology is advantageous for mechanistic comparisons. However, I have some concerns about the limitations in auditory nerve (AN) physiology. Due to practical constraints, it is not feasible to record from a large enough population of fibers that covers a full range of best frequencies (BFs) and spontaneous rates (SRs) within each animal. This raises questions about how representative the physiological data are for understanding the mechanism in behavioral data. I am curious about the authors' interpretation of how this stimulus setup might influence results compared to methods used by Kale and Heinz (2010), who adjusted harmonic frequencies based on the characteristic frequency (CF) of recorded units. While, the harmonic frequencies in this study are fixed across all CFs, meaning that many AN fibers may not be tuned closely to the stimulus frequencies. If units are not responsive to the stimulus further clarification on detecting mistuning and phase locking to TFS effects within this setup would be valuable. Given the limited number of units per condition-sometimes as few as three for certain conditions-I wonder if CF-dependent variability might impact the results of the AN data in this study and discussing this factor can help with better understanding the results. While the use of the same stimuli for both behavioral and physiological recordings is understandable, a discussion on how this choice affects interpretation would be beneficial. In addition a 60 dB stimulus could saturate high spontaneous rate (HSR) AN fibers, influencing neural coding and phase-locking to TFS. Potentially separating SR groups, could help address these issues and improve interpretive clarity.

A deeper discussion on the role of fiber spontaneous rate could also enhance the study. How might considering SR groups affect AN results related to TFS coding? While some statistical measures are included in the supplement, a more detailed discussion in the main text could help in interpretation.

Although Figure S2 indicates no change in median SR, the high-dose treatment group lacks LSR fibers, suggesting a different distribution based on SR for different animal groups, as seen in similar studies on other species. A histogram of these results would be informative, as LSR fiber loss with CS-whether induced by ouabain in gerbils or noise in other animals-is well documented (e.g., Furman et al., 2013).

Although ouabain effects on gerbils have been explored in previous studies, since these data is already seems to be recorded for the animal in this study, a brief description of changes in auditory brainstem response (ABR) thresholds, wave 1 amplitudes, and tuning curves for animals with cochlear synaptopathy (CS) in this study would be beneficial. This would confirm that ouabain selectively affects synapses without impacting outer hair cells (OHCs). For aged animals, since ABR measurements were taken, comparing hearing differences between normal and aged groups could provide insights into the pathologies besides CS in aged animals. Additionally, examining subject variability in treatment effects on hearing and how this correlates with behavior and physiology would yield valuable insights. If limited space maybe a brief clarification or inclusion in supplementary could be good enough.

Another suggestion is to discuss the potential role of MOC efferent system and effect of anesthesia in reducing efferent effects in AN recordings. This is particularly relevant for aged animals, as CS might affect LSR fibers, potentially disrupting the medial olivocochlear (MOC) efferent pathway. Anesthesia could lessen MOC activity in both young and aged animals, potentially masking efferent effects that might be present in behavioral tasks. Young gerbils with functional efferent systems might perform better behaviorally, while aged gerbils with impaired MOC function due to CS might lack this advantage. A brief discussion on this aspect could potentially enhance mechanistic insights.

Lastly, although synapse counts did not differ between the low-dose treatment and NH I sham groups, separating these groups rather than combining them with the sham might reveal differences in behavior or AN results, particularly regarding the significance of differences between aged/treatment groups and the young normal-hearing group.

---

## [Referee Report · Reviewer #2 (Public review)]

Summary:

Using a gerbil model, the authors tested the hypothesis that loss of synapses between sensory hair cells and auditory nerve fibers (which may occur due to noise exposure or aging) affects behavioral discrimination of the rapid temporal fluctuations of sounds. In contrast to previous suggestions in the literature, their results do not support this hypothesis; young animals treated with a compound that reduces the number of synapses did not show impaired discrimination compared to controls. Additionally, their results from older animals showing impaired discrimination suggest that age-related changes aside from synaptopathy are responsible for the age-related decline in discrimination.

Strengths:

(1) The rationale and hypothesis are well-motivated and clearly presented.

(2) The study was well conducted with strong methodology for the most part, and good experimental control. The combination of physiological and behavioral techniques is powerful and informative. Reducing synapse counts fairly directly using ouabain is a cleaner design than using noise exposure or age (as in other studies), since these latter modifiers have additional effects on auditory function.

(3) The study may have a considerable impact on the field. The findings could have important implications for our understanding of cochlear synaptopathy, one of the most highly researched and potentially impactful developments in hearing science in the past fifteen years.

Weaknesses:

(1) I have concerns that the gerbils may not have been performing the behavioral task using temporal fine structure information.

Human studies using the same task employed a filter center frequency that was (at least) 11 times the fundamental frequency (Marmel et al., 2015; Moore and Sek, 2009). Moore and Sek wrote: "the default (recommended) value of the centre frequency is 11F0." Here, the center frequency was only 4 or 8 times the fundamental frequency (4F0 or 8F0). Hence, relative to harmonic frequency, the harmonic spacing was considerably greater in the present study. However, gerbil auditory filters are thought to be broader than those in human. In the revised version of the manuscript, the authors provide modelling results suggesting that the excitation patterns were discriminable for the 4F0 conditions, but may not have been for the 8F0 conditions. These results provide some reassurance that the 8F0 discriminations were dependent on temporal cues, but the description of the model lacks detail. Also, the authors state that "thus, for these two conditions with harmonic number N of 8 the gerbils cannot rely on differences in the excitation patterns but must solve the task by comparing the temporal fine structure." This is too strong. Pulsed tone intensity difference limens (the reference used for establishing whether or not the excitation pattern cues were usable) may not be directly comparable to profile-analysis-like conditions, and it has been argued that frequency discrimination may be more sensitive to excitation pattern cues than predicted from a simple comparison to intensity difference limens (Micheyl et al. 2013, https://doi.org/10.1371/journal.pcbi.1003336).

I'm also somewhat concerned that the masking noise used in the present study was too low in level to mask cochlear distortion products. Based on their excitation pattern modelling, the authors state (without citation) that "since the level of excitation produced by the pink noise is less than 30 dB below that produced by the complex tones, distortion products will be masked." The basis for this claim is not clear. In human, distortion products may be only ~20 dB below the levels of the primaries (referenced to an external sound masker / canceller, which is appropriate, assuming that the modelling reported in the present paper did not include middle-ear effects; see Norman-Haignere and McDermott, 2016, doi: 10.1016/j.neuroimage.2016.01.050). Oxenham et al. (2009, doi: 10.1121/1.3089220) provide further cautionary evidence on the potential use of distortion product cues when the background noise level is too low (in their case the relative level of the noise in the compromised condition was only a little below that used in the present study). The masking level used in the present study may have been sufficient, but it would be useful to have some further reassurance on this point.

(2) The synapse reductions in the high ouabain and old groups were relatively small (mean of 19 synapses per hair cell compared to 23 in the young untreated group). In contrast, in some mouse models of the effects of noise exposure or age, a 50% reduction in synapses is observed, and in the human temporal bone study of Wu et al. (2021, https://doi.org/10.1523/JNEUROSCI.3238-20.2021) the age-related reduction in auditory nerve fibres was ~50% or greater for the highest age group across cochlear location. It could be simply that the synapse loss in the present study was too small to produce significant behavioral effects. Hence, although the authors provide evidence that in the gerbil model the age-related behavioral effects are not due to synaptopathy, this may not translate to other species (including human).

(3) The study was not pre-registered, and there was no a priori power calculation, so there is less confidence in replicability than could have been the case. Only three old animals were used in the behavioral study, which raises concerns about the reliability of comparisons involving this group. Statistical analyses on very small samples can be unreliable due to problems of power, generalisability, and susceptibility to outliers.

---

## [Referee Report · Reviewer #3 (Public review)]

This study is a part of the ongoing series of rigorous work from this group exploring neural coding deficits in the auditory nerve, and dissociating the effects of cochlear synaptopathy from other age-related deficits. They have previously shown no evidence of phase-locking deficits in the remaining auditory nerve fibers in quiet-aged gerbils. Here, they study the effects of aging on the perception and neural coding of temporal fine structure cues in the same Mongolian gerbil model.

They measure TFS coding in the auditory nerve using the TFS1 task which uses a combination of harmonic and tone-shifted inharmonic tones which differ primarily in their TFS cues (and not the envelope). They then follow this up with a behavioral paradigm using the TFS1 task in these gerbils. They test young normal hearing gerbils, aged gerbils, and young gerbils with cochlear synaptopathy induced using the neurotoxin ouabain to mimic synapse losses seen with age.

In the behavioral paradigm, they find that aging is associated with decreased performance compared to the young gerbils, whereas young gerbils with similar levels of synapse loss do not show these deficits. When looking at the auditory nerve responses, they find no differences in neural coding of TFS cues across any of the groups. However, aged gerbils show an increase in the representation of periodicity envelope cues (around f0) compared to young gerbils or those with induced synapse loss. The authors hence conclude that synapse loss by itself doesn't seem to be important for distinguishing TFS cues, and rather the behavioral deficits with age are likely having to do with the misrepresented envelope cues instead.

The manuscript is well written, and the data presented are robust. Some of the points below will need to be considered while interpreting the results of the study, in its current form. These considerations are addressable if deemed necessary, with some additional analysis in future versions of the manuscript.

Spontaneous rates - Figure S2 shows no differences in median spontaneous rates across groups. But taking the median glosses over some of the nuances there. Ouabain (in the Bourien study) famously affects low spont rates first, and at a higher degree than median or high spont rates. It seems to be the case (qualitatively) in figure S2 as well, with almost no units in the low spont region in the ouabain group, compared to the other groups. Looking at distributions within each spont rate category and comparing differences across the groups might reveal some of the underlying causes for these changes. Given that overall, the study reports that low-SR fibers had a higher ENV/TFS log-z-ratio, the distribution of these fibers across groups may reveal specific effects of TFS coding by group.

[Update: The revised manuscript has addressed these issues]

Threshold shifts - It is unclear from the current version if the older gerbils have changes in hearing thresholds, and whether those changes may be affecting behavioral thresholds. The behavioral stimuli appear to have been presented at a fixed sound level for both young and aged gerbils, similar to the single unit recordings. Hence, age-related differences in behavior may have been due to changes in relative sensation level. Approaches such as using hearing thresholds as covariates in the analysis will help explore if older gerbils still show behavioral deficits.

[Update: The issue of threshold shifts with aging gerbils is still unresolved in my opinion. From the revised manuscript, it appears that aged gerbils have a 36dB shift in thresholds. While the revised manuscript provides convincing evidence that these threshold shifts do not affect the auditory nerve tuning properties, the behavioral paradigm was still presented at the same sound level for young and aged animals. But a potential 36 dB change in sensation level may affect behavioral results. The authors may consider adding thresholds as covariates in analyses or present any evidence that behavioral thresholds are plateaued along that 30dB range].

Task learning in aged gerbils - It is unclear if the aged gerbils really learn the task well in two of the three TFS1 test conditions. The d' of 1 which is usually used as the criterion for learning was not reached in even the easiest condition for aged gerbils in all but one condition for the aged gerbils (Fig. 5H) and in that condition, there doesn't seem to be any age-related deficits in behavioral performance (Fig. 6B). Hence dissociating the inability to learn the task from the inability to perceive TFS 1 cues in those animals becomes challenging.

[Update: The revised manuscript sufficiently addresses these issues, with the caveat of hearing threshold changes affecting behavioral thresholds mentioned above].

Increased representation of periodicity envelope in the AN - the mechanisms for increased representation of periodicity envelope cues is unclear. The authors point to some potential central mechanisms but given that these are recordings from the auditory nerve what central mechanisms these may be is unclear. If the authors are suggesting some form of efferent modulation only at the f0 frequency, no evidence for this is presented. It appears more likely that the enhancement may be due to outer hair cell dysfunction (widened tuning, distorted tonotopy). Given this increased envelope coding, the potential change in sensation level for the behavior (from the comment above), and no change in neural coding of TFS cues across any of the groups, a simpler interpretation may be -TFS coding is not affected in remaining auditory nerve fibers after age-related or ouabain induced synapse loss, but behavioral performance is affected by altered outer hair cell dysfunction with age.

[Update: The revised manuscript has addressed these issues]

Emerging evidence seems to suggest that cochlear synaptopathy and/or TFS encoding abilities might be reflected in listening effort rather than behavioral performance. Measuring some proxy of listening effort in these gerbils (like reaction time) to see if that has changed with synapse loss, especially in the young animals with induced synaptopathy, would make an interesting addition to explore perceptual deficits of TFS coding with synapse loss.

[Update: The revised manuscript has addressed these issues]

---

## [Author Response]

The following is the authors’ response to the current reviews.

**Reviewer #2 (Public review):**
Summary:Using a gerbil model, the authors tested the hypothesis that loss of synapses between sensory hair cells and auditory nerve fibers (which may occur due to noise exposure or aging) affects behavioral discrimination of the rapid temporal fluctuations of sounds. In contrast to previous suggestions in the literature, their results do not support this hypothesis; young animals treated with a compound that reduces the number of synapses did not show impaired discrimination compared to controls. Additionally, their results from older animals showing impaired discrimination suggest that age-related changes aside from synaptopathy are responsible for the age-related decline in discrimination.Strengths:(1) The rationale and hypothesis are well-motivated and clearly presented.(2) The study was well conducted with strong methodology for the most part, and good experimental control. The combination of physiological and behavioral techniques is powerful and informative. Reducing synapse counts fairly directly using ouabain is a cleaner design than using noise exposure or age (as in other studies), since these latter modifiers have additional effects on auditory function.(3) The study may have a considerable impact on the field. The findings could have important implications for our understanding of cochlear synaptopathy, one of the most highly researched and potentially impactful developments in hearing science in the past fifteen years.Weaknesses:(1) I have concerns that the gerbils may not have been performing the behavioral task using temporal fine structure information.Human studies using the same task employed a filter center frequency that was (at least) 11 times the fundamental frequency (Marmel et al., 2015; Moore and Sek, 2009). Moore and Sek wrote: "the default (recommended) value of the centre frequency is 11F0." Here, the center frequency was only 4 or 8 times the fundamental frequency (4F0 or 8F0). Hence, relative to harmonic frequency, the harmonic spacing was considerably greater in the present study. However, gerbil auditory filters are thought to be broader than those in human. In the revised version of the manuscript, the authors provide modelling results suggesting that the excitation patterns were discriminable for the 4F0 conditions, but may not have been for the 8F0 conditions. These results provide some reassurance that the 8F0 discriminations were dependent on temporal cues, but the description of the model lacks detail. Also, the authors state that "thus, for these two conditions with harmonic number N of 8 the gerbils cannot rely on differences in the excitation patterns but must solve the task by comparing the temporal fine structure." This is too strong. Pulsed tone intensity difference limens (the reference used for establishing whether or not the excitation pattern cues were usable) may not be directly comparable to profile-analysis-like conditions, and it has been argued that frequency discrimination may be more sensitive to excitation pattern cues than predicted from a simple comparison to intensity difference limens (Micheyl et al. 2013, https://doi.org/10.1371/journal.pcbi.1003336)

We can assume that our conclusions based on the excitation patterns are adequate when putting gerbil auditory filter data, frequency difference limens and intensity difference limens together into perspective. Kittel et al. (2002) observed an about factor 2 larger auditory-filter bandwidth in the gerbil than in humans reducing the number of independent frequency channels in the analysis of excitation patterns. The gerbil frequency-difference limen for pure tones being an indicator for the sensitivity to make use of excitation patterns is more than an order of magnitude larger than the corresponding human frequency difference limen (Klinge and Klump 2009, https://doi.org/10.1121/1.3021315). Finally, the gerbil intensity-difference limen of 2.8 dB observed for 1-kHz pure tones is considerably larger than the 0.75 dB observed for humans in the same study (Sinnott et al. 1992). Thus, taken together these lines of evidence indicate that our conclusions regarding the potential use of excitation patterns are not too strong.

I'm also somewhat concerned that the masking noise used in the present study was too low in level to mask cochlear distortion products. Based on their excitation pattern modelling, the authors state (without citation) that "since the level of excitation produced by the pink noise is less than 30 dB below that produced by the complex tones, distortion products will be masked." The basis for this claim is not clear. In human, distortion products may be only ~20 dB below the levels of the primaries (referenced to an external sound masker / canceller, which is appropriate, assuming that the modelling reported in the present paper did not include middle-ear effects; see Norman-Haignere and McDermott, 2016, doi: 10.1016/j.neuroimage.2016.01.050). Oxenham et al. (2009, doi: 10.1121/1.3089220) provide further cautionary evidence on the potential use of distortion product cues when the background noise level is too low (in their case the relative level of the noise in the compromised condition was only a little below that used in the present study). The masking level used in the present study may have been sufficient, but it would be useful to have some further reassurance on this point.

In the method section, we provide the citation for estimating the size of the distortion products and the estimated signal-to-noise ratio making the basis for our estimates clear.

We consulted Oxenham et al. (2009, doi: 10.1121/1.3089220) who suggested that distortion products may have been used in human subjects. However, in Fig. 1 of their paper, they convincingly demonstrate that even for humans that have more narrow auditory filters than gerbils, spectral cues cannot be used to evaluate the frequency shift in harmonic complex tones. We are confident that the same limitation applies to gerbils that have wider auditory filters than humans and a lower ability to use spectral cues as indicated by their higher frequency-difference limens and intensity-difference limens compared to humans.

(2) The synapse reductions in the high ouabain and old groups were relatively small (mean of 19 synapses per hair cell compared to 23 in the young untreated group). In contrast, in some mouse models of the effects of noise exposure or age, a 50% reduction in synapses is observed, and in the human temporal bone study of Wu et al. (2021, https://doi.org/10.1523/JNEUROSCI.3238-20.2021) the age-related reduction in auditory nerve fibres was ~50% or greater for the highest age group across cochlear location. It could be simply that the synapse loss in the present study was too small to produce significant behavioral effects. Hence, although the authors provide evidence that in the gerbil model the age-related behavioral effects are not due to synaptopathy, this may not translate to other species (including human).(3) The study was not pre-registered, and there was no a priori power calculation, so there is less confidence in replicability than could have been the case. Only three old animals were used in the behavioral study, which raises concerns about the reliability of comparisons involving this group.
**Reviewer #3 (Public review):**
This study is a part of the ongoing series of rigorous work from this group exploring neural coding deficits in the auditory nerve, and dissociating the effects of cochlear synaptopathy from other age-related deficits. They have previously shown no evidence of phase-locking deficits in the remaining auditory nerve fibers in quiet-aged gerbils. Here, they study the effects of aging on the perception and neural coding of temporal fine structure cues in the same Mongolian gerbil model.They measure TFS coding in the auditory nerve using the TFS1 task which uses a combination of harmonic and tone-shifted inharmonic tones which differ primarily in their TFS cues (and not the envelope). They then follow this up with a behavioral paradigm using the TFS1 task in these gerbils. They test young normal hearing gerbils, aged gerbils, and young gerbils with cochlear synaptopathy induced using the neurotoxin ouabain to mimic synapse losses seen with age.In the behavioral paradigm, they find that aging is associated with decreased performance compared to the young gerbils, whereas young gerbils with similar levels of synapse loss do not show these deficits. When looking at the auditory nerve responses, they find no differences in neural coding of TFS cues across any of the groups. However, aged gerbils show an increase in the representation of periodicity envelope cues (around f0) compared to young gerbils or those with induced synapse loss. The authors hence conclude that synapse loss by itself doesn't seem to be important for distinguishing TFS cues, and rather the behavioral deficits with age are likely having to do with the misrepresented envelope cues instead.The manuscript is well written, and the data presented are robust. Some of the points below will need to be considered while interpreting the results of the study, in its current form. These considerations are addressable if deemed necessary, with some additional analysis in future versions of the manuscript.Spontaneous rates - Figure S2 shows no differences in median spontaneous rates across groups. But taking the median glosses over some of the nuances there. Ouabain (in the Bourien study) famously affects low spont rates first, and at a higher degree than median or high spont rates. It seems to be the case (qualitatively) in figure S2 as well, with almost no units in the low spont region in the ouabain group, compared to the other groups. Looking at distributions within each spont rate category and comparing differences across the groups might reveal some of the underlying causes for these changes. Given that overall, the study reports that low-SR fibers had a higher ENV/TFS log-z-ratio, the distribution of these fibers across groups may reveal specific effects of TFS coding by group.[Update: The revised manuscript has addressed these issues]Threshold shifts - It is unclear from the current version if the older gerbils have changes in hearing thresholds, and whether those changes may be affecting behavioral thresholds. The behavioral stimuli appear to have been presented at a fixed sound level for both young and aged gerbils, similar to the single unit recordings. Hence, age-related differences in behavior may have been due to changes in relative sensation level. Approaches such as using hearing thresholds as covariates in the analysis will help explore if older gerbils still show behavioral deficits.[Update: The issue of threshold shifts with aging gerbils is still unresolved in my opinion. From the revised manuscript, it appears that aged gerbils have a 36dB shift in thresholds. While the revised manuscript provides convincing evidence that these threshold shifts do not affect the auditory nerve tuning properties, the behavioral paradigm was still presented at the same sound level for young and aged animals. But a potential 36 dB change in sensation level may affect behavioral results. The authors may consider adding thresholds as covariates in analyses or present any evidence that behavioral thresholds are plateaued along that 30dB range].

Since we do not have behavioural detection thresholds from our individual animals, only CAP thresholds that represent the auditory-nerve data and cannot be translated to behavioural thresholds directly, we want to refrain from using these indirect measures as covariates in the present analysis. In addition, the study by Hamann et al. (2002, https://doi.org/10.1016/S0378-5955(02)00454-9) indicates that age-related behavioural threshold increases are smaller than threshold increases obtained from auditory brainstem response measurements. Finally, statistical analyses on very small samples can be unreliable due to problems of power, generalisability, and susceptibility to outliers.

Moore and Sek (2009) in their paper on the TFS1 test pointed out that the effect of signal level on the TFS1 threshold in normal hearing human subjects was small when the signal-to-noise ratio between the broadband masking noise and the complex tone was kept constant. Furthermore, the masking noise will raise the thresholds of normal hearing gerbils and old gerbils with an audibility threshold increase to about the same signal-to-noise ratio. Thus, as long as the signal remains audible to the behaviourally tested gerbil which can be expected at an overall signal level of 68 dB SPL, we expect little effect of raised audibility thresholds on the TFS1 threshold. The lack of temporal processing deficits in the auditory-nerve fibers of old, mildly hearing impaired gerbils compared to those in normal hearing young adult gerbils further strengthens this argument.

Task learning in aged gerbils - It is unclear if the aged gerbils really learn the task well in two of the three TFS1 test conditions. The d' of 1 which is usually used as the criterion for learning was not reached in even the easiest condition for aged gerbils in all but one condition for the aged gerbils (Fig. 5H) and in that condition, there doesn't seem to be any age-related deficits in behavioral performance (Fig. 6B). Hence dissociating the inability to learn the task from the inability to perceive TFS 1 cues in those animals becomes challenging.[Update: The revised manuscript sufficiently addresses these issues, with the caveat of hearing threshold changes affecting behavioral thresholds mentioned above].

As we argued above, an audibility threshold increase in the old gerbils is unlikely to explain the raised TFS1 thresholds in the old gerbils.

Increased representation of periodicity envelope in the AN - the mechanisms for increased representation of periodicity envelope cues is unclear. The authors point to some potential central mechanisms but given that these are recordings from the auditory nerve what central mechanisms these may be is unclear. If the authors are suggesting some form of efferent modulation only at the f0 frequency, no evidence for this is presented. It appears more likely that the enhancement may be due to outer hair cell dysfunction (widened tuning, distorted tonotopy). Given this increased envelope coding, the potential change in sensation level for the behavior (from the comment above), and no change in neural coding of TFS cues across any of the groups, a simpler interpretation may be -TFS coding is not affected in remaining auditory nerve fibers after age-related or ouabain induced synapse loss, but behavioral performance is affected by altered outer hair cell dysfunction with age.[Update: The revised manuscript has addressed these issues]Emerging evidence seems to suggest that cochlear synaptopathy and/or TFS encoding abilities might be reflected in listening effort rather than behavioral performance. Measuring some proxy of listening effort in these gerbils (like reaction time) to see if that has changed with synapse loss, especially in the young animals with induced synaptopathy, would make an interesting addition to explore perceptual deficits of TFS coding with synapse loss.[Update: The revised manuscript has addressed these issues]
**Reviewer #3 (Recommendations for the authors):**
Thank you for your revisions. They largely address most of my initial concerns. The issue of threshold shifts potentially affecting behavioral thresholds still remains unresolved in my opinion. The new data about unaltered tuning curves is convincing that the auditory nerve fiber recordings are unaffected by threshold shifts. But am I correct in my understanding that the threshold shift with age was 36 dB relative to the young (L168)? If so, wouldn't the fact that behavior was performed at 68 dB SPL regardless of group affect the behavioral thresholds with age? Is there any additional evidence that suggests that behavioral performance plateaus along that ~30dB range that the authors could include to strengthen this claim?

In our response above to reviewer #3 and to reviewer #2 we provided additional arguments why we think that an audibility threshold increase in old gerbils cannot explain their compromised TFS1 thresholds.

The following is the authors’ response to the original reviews.

**Reviewer #1(Public review)**
Summary:The authors investigate the effects of aging on auditory system performance in understanding temporal fine structure (TFS), using both behavioral assessments and physiological recordings from the auditory periphery, specifically at the level of the auditory nerve. This dual approach aims to enhance understanding of the mechanisms underlying observed behavioral outcomes. The results indicate that aged animals exhibit deficits in behavioral tasks for distinguishing between harmonic and inharmonic sounds, which is a standard test for TFS coding. However, neural responses at the auditory nerve level do not show significant differences when compared to those in young, normalhearing animals. The authors suggest that these behavioral deficits in aged animals are likely attributable to dysfunctions in the central auditory system, potentially as a consequence of aging. To further investigate this hypothesis, the study includes an animal group with selective synaptic loss between inner hair cells and auditory nerve fibers, a condition known as cochlear synaptopathy (CS).CS is a pathology associated with aging and is thought to be an early indicator of hearing impairment. Interestingly, animals with selective CS showed physiological and behavioral TFS coding similar to that of the young normal-hearing group, contrasting with the aged group's deficits. Despite histological evidence of significant synaptic loss in the CS group, the study concludes that CS does not appear to affect TFS coding, either behaviorally or physiologically.

We agree with the reviewer’s summary.

Strengths:This study addresses a critical health concern, enhancing our understanding of mechanisms underlying age-related difficulties in speech intelligibility, even when audiometric thresholds are within normal limits. A major strength of this work is the comprehensive approach, integrating behavioral assessments, auditory nerve (AN) physiology, and histology within the same animal subjects. This approach enhances understanding of the mechanisms underlying the behavioral outcomes and provides confidence in the actual occurrence of synapse loss and its effects. The study carefully manages controlled conditions by including five distinct groups: young normal-hearing animals, aged animals, animals with CS induced through low and high doses, and a sham surgery group. This careful setup strengthens the study's reliability and allows for meaningful comparisons across conditions. Overall, the manuscript is well-structured, with clear and accessible writing that facilitates comprehension of complex concepts.Weaknesses:The stimulus and task employed in this study are very helpful for behavioral research, and using the same stimulus setup for physiology is advantageous for mechanistic comparisons. However, I have some concerns about the limitations in auditory nerve (AN) physiology. Due to practical constraints, it is not feasible to record from a large enough population of fibers that covers a full range of best frequencies (BFs) and spontaneous rates (SRs) within each animal. This raises questions about how representative the physiological data are for understanding the mechanism in behavioral data. I am curious about the authors' interpretation of how this stimulus setup might influence results compared to methods used by Kale and Heinz (2010), who adjusted harmonic frequencies based on the characteristic frequency (CF) of recorded units. While, the harmonic frequencies in this study are fixed across all CFs, meaning that many AN fibers may not be tuned closely to the stimulus frequencies. If units are not responsive to the stimulus further clarification on detecting mistuning and phase locking to TFS effects within this setup would be valuable. Since the harmonic frequencies in this study are fixed across all CFs, this means that many AN fibers may not be tuned closely to the stimulus frequencies, adding sampling variability to the results.

We chose the stimuli for the AN recordings to be identical to the stimuli used in the behavioral evaluation of the perceptual sensitivity. Only with this approach can we directly compare the response of the population of AN fibers with perception measured in behavior.

The stimuli are complex, i.e., comprise of many frequency components AND were presented at 68 dB SPL. Thus, the stimuli excite a given fiber within a large portion of the fiber’s receptive field. Furthermore, during recordings, we assured ourselves that fibers responded to the stimuli by audiovisual control. Otherwise it would have cost valuable recording time to record from a nonresponsive AN fiber.

Given the limited number of units per condition-sometimes as few as three for certain conditions - I wonder if CF-dependent variability might impact the results of the AN data in this study and discussing this factor can help with better understanding the results. While the use of the same stimuli for both behavioral and physiological recordings is understandable, a discussion on how this choice affects interpretation would be beneficial. In addition a 60 dB stimulus could saturate high spontaneous rate (HSR) AN fibers, influencing neural coding and phase-locking to TFS. Potentially separating SR groups, could help address these issues and improve interpretive clarity.A deeper discussion on the role of fiber spontaneous rate could also enhance the study. How might considering SR groups affect AN results related to TFS coding? While some statistical measures are included in the supplement, a more detailed discussion in the main text could help in interpretation. We do not think that it will be necessary to conduct any statistical analysis in addition to that already reported in the supplement.

We considered moving some supplementary information back into the main manuscript but decided against it. Our single-unit sample was not sufficient, i.e. not all subpopulations of auditory-nerve fibers were sufficiently sampled for all animal treatment groups, to conclusively resolve every aspect that may be interesting to explore. The power of our approach lies in the direct linkage of several levels of investigation – cochlear synaptic morphology, single-unit representation and behavioral performance – and, in the main manuscript, we focus on the core question of synaptopathy and its relation to temporal fine structure perception. This is now spelled out clearly in lines 197 - 203 of the main manuscript.

Although Figure S2 indicates no change in median SR, the high-dose treatment group lacks LSR fibers, suggesting a different distribution based on SR for different animal groups, as seen in similar studies on other species. A histogram of these results would be informative, as LSR fiber loss with CS-whether induced by ouabain in gerbils or noise in other animals-is well documented (e.g., Furman et al., 2013).

Figure S2 was revised to avoid overlap of data points and show the distributions more clearly. Furthermore, the sample sizes for LSR and HSR fibers are now provided separately.

Although ouabain effects on gerbils have been explored in previous studies, since these data already seems to be recorded for the animal in this study, a brief description of changes in auditory brainstem response (ABR) thresholds, wave 1 amplitudes, and tuning curves for animals with cochlear synaptopathy (CS) in this study would be beneficial. This would confirm that ouabain selectively affects synapses without impacting outer hair cells (OHCs). For aged animals, since ABR measurements were taken, comparing hearing differences between normal and aged groups could provide insights into the pathologies besides CS in aged animals. Additionally, examining subject variability in treatment effects on hearing and how this correlates with behavior and physiology would yield valuable insights. If limited space maybe a brief clarification or inclusion in supplementary could be good enough.

We thank the reviewer for this constructive suggestion. The requested data were added in a new section of the Results, entitled “Threshold sensitivity and frequency tuning were not affected by the synapse loss.” (lines 150 – 174). Our young-adult, ouabain-treated gerbils showed no significant elevations of CAP thresholds and their neural tuning was normal. Old gerbils showed the typical threshold losses for individuals of comparable age, and normal neural tuning, confirming previous reports. Thus, there was no evidence for relevant OHC impairments in any of our animal groups.

Another suggestion is to discuss the potential role of MOC efferent system and effect of anesthesia in reducing efferent effects in AN recordings. This is particularly relevant for aged animals, as CS might affect LSR fibers, potentially disrupting the medial olivocochlear (MOC) efferent pathway. Anesthesia could lessen MOC activity in both young and aged animals, potentially masking efferent effects that might be present in behavioral tasks. Young gerbils with functional efferent systems might perform better behaviorally, while aged gerbils with impaired MOC function due to CS might lack this advantage. A brief discussion on this aspect could potentially enhance mechanistic insights.

Thank you for this suggestion. The potential role of olivocochlear efferents is now discussed in lines 597 - 613.

Lastly, although synapse counts did not differ between the low-dose treatment and NH I sham groups, separating these groups rather than combining them with the sham might reveal differences in behavior or AN results, particularly regarding the significance of differences between aged/treatment groups and the young normal-hearing group.

For maximizing statistical power, we combined those groups in the statistical analysis. These two groups did not differ in synapse number, threshold sensitivity or neural tuning bandwidths.

**Reviewer #2 (Public review):**
Summary:Using a gerbil model, the authors tested the hypothesis that loss of synapses between sensory hair cells and auditory nerve fibers (which may occur due to noise exposure or aging) affects behavioral discrimination of the rapid temporal fluctuations of sounds. In contrast to previous suggestions in the literature, their results do not support this hypothesis; young animals treated with a compound that reduces the number of synapses did not show impaired discrimination compared to controls. Additionally, their results from older animals showing impaired discrimination suggest that agerelated changes aside from synaptopathy are responsible for the age-related decline in discrimination.

We agree with the reviewer’s summary.

Strengths:(1) The rationale and hypothesis are well-motivated and clearly presented.(2) The study was well conducted with strong methodology for the most part, and good experimental control. The combination of physiological and behavioral techniques is powerful and informative. Reducing synapse counts fairly directly using ouabain is a cleaner design than using noise exposure or age (as in other studies), since these latter modifiers have additional effects on auditory function.(3) The study may have a considerable impact on the field. The findings could have important implications for our understanding of cochlear synaptopathy, one of the most highly researched and potentially impactful developments in hearing science in the past fifteen years.Weaknesses:(1) My main concern is that the stimuli may not have been appropriate for assessing neural temporal coding behaviorally. Human studies using the same task employed a filter center frequency that was (at least) 11 times the fundamental frequency (Marmel et al., 2015; Moore and Sek, 2009). Moore and Sek wrote: "the default (recommended) value of the centre frequency is 11F0." Here, the center frequency was only 4 or 8 times the fundamental frequency (4F0 or 8F0). Hence, relative to harmonic frequency, the harmonic spacing was considerably greater in the present study. By my calculations, the masking noise used in the present study was also considerably lower in level relative to the harmonic complex than that used in the human studies. These factors may have allowed the animals to perform the task using cues based on the pattern of activity across the neural array (excitation pattern cues), rather than cues related to temporal neural coding. The authors show that mean neural driven rate did not change with frequency shift, but I don't understand the relevance of this. It is the change in response of individual fibers with characteristic frequencies near the lowest audible harmonic that is important here.

The auditory filter bandwidth of the gerbil is about double that of human subjects. Because of this, the masking noise has a larger overall level than in the human studies in the filter, prohibiting the use of distortion products. The larger auditory filter bandwidth precludes that the gerbils can use excitation patterns, especially in the condition with a center frequency of 1600 Hz and a fundamental of 200 Hz and in the condition with a center frequency of 3200 Hz and a fundamental of 400 Hz. In the condition with a center frequency of 1600 Hz and a fundamental of 400 Hz, it is possible that excitation patterns are exploited. We have now added modeling of the excitation patterns, and a new figure showing their change at the gerbils’ perception threshold, in the discussion of the revised version (lines 440 - 446 and Fig. 8).

The case against excitation pattern cues needs to be better made in the Discussion. It could be that gerbil frequency selectivity is broad enough for this not to be an issue, but more detail needs to be provided to make this argument. The authors should consider what is the lowest audible harmonic in each case for their stimuli, given the level of each harmonic and the level of the pink noise. Even for the 8F0 center frequency, the lowest audible harmonic may be as low as the 4th (possibly even the 3rd). In human, harmonics are thought to be resolvable by the cochlea up to at least the 8th.

This issue is now covered in the discussion, see response to the previous point.

(2) The synapse reductions in the high ouabain and old groups were relatively small (mean of 19 synapses per hair cell compared to 23 in the young untreated group). In contrast, in some mouse models of the effects of noise exposure or age, a 50% reduction in synapses is observed, and in the human temporal bone study of Wu et al. (2021, https://doi.org/10.1523/JNEUROSCI.3238-20.2021) the age-related reduction in auditory nerve fibres was ~50% or greater for the highest age group across cochlear location. It could be simply that the synapse loss in the present study was too small to produce significant behavioral effects. Hence, although the authors provide evidence that in the gerbil model the age-related behavioral effects are not due to synaptopathy, this may not translate to other species (including human). This should be discussed in the manuscript.

We agree that our results apply to moderate synaptopathy, which predominantly characterizes early stages of hearing loss or aged individuals without confounding noise-induced cochlear damage. This is now discussed in lines 486 – 498.

It would be informative to provide synapse counts separately for the animals who were tested behaviorally, to confirm that the pattern of loss across the group was the same as for the larger sample.

Yes, the pattern was the same for the subgroup of behaviorally tested animals. We have added this information to the revised version of the manuscript (lines 137 – 141).

(3) The study was not pre-registered, and there was no a priori power calculation, so there is less confidence in replicability than could have been the case. Only three old animals were used in the behavioral study, which raises concerns about the reliability of comparisons involving this group.

The results for the three old subjects differed significantly from those of young subjects and young ouabain-treated subjects. This indicates a sufficient statistical power, since otherwise no significant differences would be observed.

**Reviewer #3 (Public review):**
This study is a part of the ongoing series of rigorous work from this group exploring neural coding deficits in the auditory nerve, and dissociating the effects of cochlear synaptopathy from other agerelated deficits. They have previously shown no evidence of phase-locking deficits in the remaining auditory nerve fibers in quiet-aged gerbils. Here, they study the effects of aging on the perception and neural coding of temporal fine structure cues in the same Mongolian gerbil model.They measure TFS coding in the auditory nerve using the TFS1 task which uses a combination of harmonic and tone-shifted inharmonic tones which differ primarily in their TFS cues (and not the envelope). They then follow this up with a behavioral paradigm using the TFS1 task in these gerbils. They test young normal hearing gerbils, aged gerbils, and young gerbils with cochlear synaptopathy induced using the neurotoxin ouabain to mimic synapse losses seen with age.In the behavioral paradigm, they find that aging is associated with decreased performance compared to the young gerbils, whereas young gerbils with similar levels of synapse loss do not show these deficits. When looking at the auditory nerve responses, they find no differences in neural coding of TFS cues across any of the groups. However, aged gerbils show an increase in the representation of periodicity envelope cues (around f0) compared to young gerbils or those with induced synapse loss. The authors hence conclude that synapse loss by itself doesn't seem to be important for distinguishing TFS cues, and rather the behavioral deficits with age are likely having to do with the misrepresented envelope cues instead.

We agree with the reviewer’s summary.

The manuscript is well written, and the data presented are robust. Some of the points below will need to be considered while interpreting the results of the study, in its current form. These considerations are addressable if deemed necessary, with some additional analysis in future versions of the manuscript.Spontaneous rates - Figure S2 shows no differences in median spontaneous rates across groups. But taking the median glosses over some of the nuances there. Ouabain (in the Bourien study) famously affects low spont rates first, and at a higher degree than median or high spont rates. It seems to be the case (qualitatively) in Figure S2 as well, with almost no units in the low spont region in the ouabain group, compared to the other groups. Looking at distributions within each spont rate category and comparing differences across the groups might reveal some of the underlying causes for these changes. Given that overall, the study reports that low-SR fibers had a higher ENV/TFS log-zratio, the distribution of these fibers across groups may reveal specific effects of TFS coding by group.

As the reviewer points out, our sample from the group treated with a high concentration of ouabain showed very few low-spontaneous-rate auditory-nerve fibers, as expected from previous work. However, this was also true, e.g., for our sample from sham-operated animals, and may thus well reflect a sampling bias. We are therefore reluctant to attach much significance to these data distributions. We now point out more clearly the limitations of our auditory-nerve sample for the exploration of interesting questions beyond our core research aim (see also response to Reviewer 1 above).

Threshold shifts - It is unclear from the current version if the older gerbils have changes in hearing thresholds, and whether those changes may be affecting behavioral thresholds. The behavioral stimuli appear to have been presented at a fixed sound level for both young and aged gerbils, similar to the single unit recordings. Hence, age-related differences in behavior may have been due to changes in relative sensation level. Approaches such as using hearing thresholds as covariates in the analysis will help explore if older gerbils still show behavioral deficits.

Unfortunately, we did not obtain behavioral thresholds that could be used here. We want to point out that the TFS 1 stimuli had an overall level of 68 dB SPL, and the pink noise masker would have increased the threshold more than expected from the moderate, age-related hearing loss in quiet. Thus, the masked thresholds for all gerbil groups are likely similar and should have no effect on the behavioral results.

Task learning in aged gerbils - It is unclear if the aged gerbils really learn the task well in two of the three TFS1 test conditions. The d' of 1 which is usually used as the criterion for learning was not reached in even the easiest condition for aged gerbils in all but one condition for the aged gerbils (Fig. 5H) and in that condition, there doesn't seem to be any age-related deficits in behavioral performance (Fig. 6B). Hence dissociating the inability to learn the task from the inability to perceive TFS 1 cues in those animals becomes challenging.

Even in the group of gerbils with the lowest sensitivity, for the condition 400/1600 the animals achieved a d’ of on average above 1. Furthermore, stimuli were well above threshold and audible, even when no discrimination could be observed. Finally, as explained in the methods, different stimulus conditions were interleaved in each session, providing stimuli that were easy to discriminate together with those being difficult to discriminate. This approach ensures that the gerbils were under stimulus control, meaning properly trained to perform the task. Thus, an inability to discriminate does not indicate a lack of proper training.

Increased representation of periodicity envelope in the AN - the mechanisms for increased representation of periodicity envelope cues is unclear. The authors point to some potential central mechanisms but given that these are recordings from the auditory nerve what central mechanisms these may be is unclear. If the authors are suggesting some form of efferent modulation only at the f0 frequency, no evidence for this is presented. It appears more likely that the enhancement may be due to outer hair cell dysfunction (widened tuning, distorted tonotopy). Given this increased envelope coding, the potential change in sensation level for the behavior (from the comment above), and no change in neural coding of TFS cues across any of the groups, a simpler interpretation may be -TFS coding is not affected in remaining auditory nerve fibers after age-related or ouabain induced synapse loss, but behavioral performance is affected by altered outer hair cell dysfunction with age.

A similar point was made by Reviewer #1. As indicated above, new data on threshold sensitivity and neural tuning were added in a new section of the Results which indirectly suggest that significant OHC pathologies were not a concern, neither in our young-adult, synaptopathic gerbils nor in the old gerbils.

Emerging evidence seems to suggest that cochlear synaptopathy and/or TFS encoding abilities might be reflected in listening effort rather than behavioral performance. Measuring some proxy of listening effort in these gerbils (like reaction time) to see if that has changed with synapse loss, especially in the young animals with induced synaptopathy, would make an interesting addition to explore perceptual deficits of TFS coding with synapse loss.

This is an interesting suggestion that we now explore in the revision of the manuscript. Reaction times can be used as a proxy for listening effort and were recorded for all responses. The the new analysis now reported in lines 378 - 396 compared young-adult control gerbils with young-adult gerbils that had been treated with the high concentration of ouabain. No differences in response latencies was found, indicating that listening effort did not change with synapse loss.

**Reviewer #1 (Recommendations for the authors):**
Figure 2: The y-axis labeled as "Frequency" is potentially misleading since there are additional frequency values on the right side of the panels. It would be helpful to clarify more in the caption what these right-side frequency values represent. Additionally, the legend could be positioned more effectively for clarity.

Thank you for your suggestion. The axis label was rephrased.

Figure 7: This figure is a bit unclear, as it appears to show two sets of gerbil data at 1500 Hz, yet the difference between them is not explained.

We added the following text to the figure legend: „The higher and lower thresholds shown for the gerbil data reflect thresholds at fc of 1600 Hz for fundamentals f0 of 200 Hz and 400 Hz, respectively.“

Maybe a short description of fmax that is used in Figure 4 could help or at least point to supplementary for finding the definition.

We thank the reviewer for pointing out this typo/inaccuracy. The correct terminology in line with the remainder of the manuscript is “fmaxpeak”. We corrected the caption of figure 5 (previously figure 4) and added the reference pointing to figure 11 (previously figure 9), which explains the terms.

I couldn't find information about the possible availability of data.

The auditory-nerve recordings reported in this paper are part of a larger study of single-unit auditorynerve responses in gerbils, formally described and published by Heeringa (2024) Single-unit data for sensory neuroscience: Responses from the auditory nerve of young-adult and aging gerbils. Scientific Data 11:411, https://doi.org/10.1038/s41597-024-03259-3. As soon as the Version of Record will be submitted, the raw single-unit data can be accessed directly through the following link: https://doi.org/10.5061/dryad.qv9s4mwn4. The data that are presented in the figures of the present manuscript and were statistically analyzed are uploaded to the Zenodo repository (https://doi.org/10.5281/zenodo.15546625).

**Reviewer #2 (Recommendations for the authors):**
L22. The term "hidden hearing loss" is used in many different ways in the literature, from being synonymous with cochlear synaptopathy, to being a description of any listening difficulties that are not accounted for by the audiogram (for which there are many other / older terms). The original usage was much more narrow than your definition here. It is not correct that Schaette and McAlpine defined HHL in the broad sense, as you imply. I suggest you avoid the term to prevent further confusion.

We eliminated the term hidden hearing loss.

L43. SNHL is undefined.

Thank you for catching that. The term is now spelled out.

L64. "whether" -> "that"

We corrected this issue.

L102. It would be informative to see the synapse counts (across groups) for the animals tested in the behavioral part of the study. Did these vary between groups in the same way?

Yes, the pattern was the same for the subgroup of behaviorally tested animals. We have added this information to the revised version of the manuscript (lines 137 – 141).

L108. How many tests were considered in the Bonferroni correction? Did this cover all reported tests in the paper?

The comparisons of synapse numbers between treatment groups were done with full Bonferroni correction, as in the other tests involving posthoc pair-wise comparisons after an ANOVA.

Figure 1 and 6 captions. Explain meaning of * and ** (criteria values).

The information was added to the figure legends of now Figs. 1 and 7.

L139. I don't follow the argument - the mean driven rate is not important. It is the rate at individual CFs and how that changes with frequency shift that provides the cue.L142. I don't follow - individual driven rates might have been a cue (some going up, some down, as frequency was shifted).

Yes, theoretically it is possible that the spectral pattern of driven rates (i.e., excitation pattern) can be specifically used for profile analysis and subsequently as a strong cue for discriminating the TFS1 stimuli. In order to shed some light on this question with regard to the actual stimuli used in this study, we added a comprehensive figure showing simulated excitation patterns (figure 8). The excitation patterns were generated with a gammatone filter bank and auditory filter bandwidths appropriate for gerbils (Kittel et al. 2002). The simulated excitation patterns allow to draw some at least semi-quantitative conclusions about the possibility of profile analysis: 1. In the 200/1600 Hz and 400/3200 Hz conditions (i.e., harmonic number of fc is 8), the difference between all inharmonic excitation patterns and the harmonic reference excitation pattern is far below the threshold for intensity discrimination (Sinnott et al. 1992). 2. In the same conditions, the statistics of the pink noise make excitation patterns differences at or beyond the filter slopes (on both high and low frequency limits) useless for frequency shift discrimination. 3. In the 400/1600 Hz condition (i.e., harmonic number of fc is 4), there is a non-negligible possibility that excitation pattern differences were a main cue for discrimination. All of these conclusions are compatible with the results of our study.

L193. Is this p-value Bonferroni corrected across the whole study? If not, the finding could well be spurious given the number of tests reported.

Yes, it is Bonferroni corrected

L330. TFS is already defined.L346. AN is already defined.L408. "temporal fine structure" -> "TFS"

It was a deliberate decision to define these terms again in the Discussion, for readers who prefer to skip most of the detailed Results.

L364-366. This argument is somewhat misleading. Cochlear resolvability largely depends on the harmonic spacing (i.e., F0) relative to harmonic frequency (in other words, on harmonic rank). Marmel et al. (2015) and Moore and Sek (2009) used a center frequency (at least) 11 times F0. Here, the center frequency was only 4 or 8 times F0. In human, this would not be sufficient to eliminate excitation pattern cues.

We have now included results from modeling the excitation patterns in the discussion with a new figure demonstrating that at a center frequency of 8 times F0, excitation patterns provide no useful cue while this is a possibility at a center frequency of 4 times F0 (Fig. 8, lines 440 - 446).

L541. Was that a spectrum level of 20 dB SPL (level per 1-Hz wide band) at 1 kHz? Need to clarify.

The power spectral density of the pink noise at 1 kHz (i.e., the level in a 1 Hz wide band centered at 1 kHz) was 13.3 dB SPL. The total level of the pink noise (including edge filters at 100 Hz and 11 kHz) was 50 dB SPL.

L919. So was the correction applied across only the tests within each ANOVA? Don't you need to control the study-wise error rate (across all primary tests) to avoid spurious findings?

We added information about the family-wise error rate (line 1077 - 1078). Since the ANOVAs tested different specific research questions, we do not think that we need to control the study-wise error rate.

**Reviewer #3 (Recommendations for the authors):**
There was no difference in TFS sensitivity in the AN fiber activity across all the groups. Potential deficits with age were only sound in the behavioral paradigm. Given that, it might make it clearer to specify that the deficits or lack thereof are in behavior, in multiple instances in the manuscript where it says synaptopathy showed no decline in TFS sensitivity (For example Line 342-344).

We carefully went through the entire text and clarified a couple more instances.

L353 - this statement is a bit too strong. It implies causality when there is only a co-occurrence of increased f0 representation and age-related behavioral deficits in TFS1 task.

The statement was rephrased as “Thus, cue representation may be associated with the perceptual deficits, but not reduced synapse numbers, as originally proposed.”

L465-467 - while this may be true, I think it is hard to say this with the current dataset where only AN fibers are being recorded from. I don't think we can say anything about afferent central mechanisms with this data set.

We agree. However, we refer here to published data on central inhibition to provide a possible explanation.

Hearing thresholds with ABRs are mentioned in the methods, but that data is not presented anywhere. Would be nice to see hearing thresholds across the various groups to account or discount outer hair cell dysfunction.

This important point was made repeatedly and we thank the Reviewers for it. As indicated above, new data on threshold sensitivity and neural tuning were added in a new section of the Results which indirectly suggest that significant OHC pathologies were not a concern, neither in our young-adult, synaptopathic gerbils nor in the old gerbils.